



# Arctic freshwater fluxes: sources, tracer budgets and inconsistencies

Alexander Forryan[1], Sheldon Bacon[2], Takamasa Tsubouchi[3],
Sinhué Torres-Valdés[4], and Alberto C. Naveira Garabato[1]

[1]Ocean and Earth Science, University of Southampton, Southampton, U.K.
[2]National Oceanography Centre, Southampton, U.K.
[3]Geophysical Institute, University of Bergen, Bergen, Norway
[4]Alfred Wegener Institute for Polar and Marine Research, Bremerhaven, Germany

Correspondence to: A. Forryan (af1c10@soton.ac.uk)





## Abstract

The traditionally divergent perspectives of the Arctic Ocean freshwater budget provided by control volume-based and geochemical tracer-based approaches are reconciled, and the sources of inter-approach inconsistencies identified, by comparing both methodologies using an observational data set of the circulation and water mass properties at the basin's boundary in summer 2005. The control volume-based and geochemical estimates of the Arctic Ocean (liquid) freshwater fluxes are $147 \pm 42$ mSv (1 Sv = $10^6$ m$^3$s$^{-1}$) and $140 \pm 67$ mSv, respectively, and are thus in agreement. Examination of meteoric, sea ice and seawater contributions to the freshwater fluxes reveals near equivalence of the net freshwater flux out of the Arctic and the meteoric source to the basin, and a close balance between the transport of solid sea ice and ice-derived meltwater out of the Arctic and the freshwater deficit in the seawater from which the sea ice has been frozen out. Inconsistencies between the two approaches are shown to stem from the distinction between "Atlantic" and "Pacific" waters based on tracers in geochemical tracer-based calculations. The definition of Pacific waters is found to be particularly problematic, because of the non-conservative nature of the inorganic nutrients underpinning that definition, as well as the low salinity characterising waters entering the Arctic through Bering Strait - which makes them difficult to isolate from meteoric sources.

## 1   Introduction

The global climate is changing (Stocker et al., 2014), and Arctic amplification is increasing both the rate and the variability of this change in the Arctic (Serreze and Barry, 2011). Despite its relatively small area, the Arctic Ocean receives a disproportionate amount of freshwater and plays a disproportionately large role in the regulation of the global climate (Carmack et al., 2016; Prowse et al., 2015). The permanent halocline, established by freshwater input into the Arctic, both promotes sea ice formation through limiting deep convection, and constrains the upward heat flux from deeper warmer waters that promotes sea





ice longevity (Carmack et al., 2016). Consequently, changes to the freshwater cycle within the Arctic potentially perturb the formation and melting of sea ice, which has in turn a pronounced impact on both the Arctic heat budget and on planetary albedo (Serreze et al., 2006; Carmack et al., 2016). Changes in the Arctic heat budget may affect the strength
of the north-south temperature gradient between the polar and mid-latitudes regions, which has recently been linked to increased probability of extreme weather events at mid-latitudes (Screen and Simmonds, 2014; Francis and Vavrus, 2012; Mann et al., 2017). Arctic freshwater export also has the potential to change Atlantic northward heat fluxes through the disruption of deep convection and, consequently, the strength of the Atlantic meridional
overturning circulation (e.g. Manabe and Stouffer, 1995).

Following the seminal work of Aagaard and Carmack (1989), the Arctic Ocean freshwater budget is usually quantified using either "direct" or "indirect" approaches (Haine et al., 2015; Serreze et al., 2006; Dickson et al., 2007; Carmack et al., 2016). The direct approach uses the net sum of river runoff, precipitation and evaporation, while the indirect approach
employs knowledge of ocean (including sea ice) salinity and volume fluxes across an Arctic boundary (Haine et al., 2015; Serreze et al., 2006; Dickson et al., 2007).

Direct measurement of Arctic freshwater fluxes is hampered by the scarcity of observations (both in-situ and remote) and incomplete knowledge and understanding of the physical processes involving air moisture, clouds, precipitation and evaporation (Vihma et al., 2016;
Bring et al., 2016; Lique et al., 2016). This scarcity is compounded by uncertainty in the observations themselves (e.g. Aleksandrov et al., 2005) and by sparsely distributed sampling sites (for a full discussion see Vihma et al., 2016). Estimates of runoff are limited by incomplete river observations (with only $\sim$ 70 % of Arctic rivers gauged) and understanding of how river discharge is modified in response to permafrost changes and subsurface
25 / surface water interactions (Bring et al., 2016, 2017). Compensation for ungauged runoff, arising from incomplete river observations, is usually achieved by the use of simple models based on linear regression from gauged regions (e.g. Shiklomanov et al., 2000; Lammers et al., 2007). The use of atmospheric reanalysis products (e.g. Haine et al., 2015) to compensate for the paucity of direct measurements is in turn hampered by the scarcity and



uncertainty of observations to constrain those reanalyses, which makes accurate modelling of all the physical processes involved problematic and leads to relatively unconstrained model dynamics in the Arctic (Lique et al., 2016).

Indirect measurement of Arctic freshwater fluxes considers the Arctic Ocean to be a basin enclosed by land boundaries and/or by ocean measurements in which inflowing ocean waters are modified by ocean surface and land boundary fluxes, both freshening and cooling, to form outflows. Therefore, with knowledge of net mass (volume) inflows and outflows combined with knowledge of a suitable tracer (in this case salinity), the net surface freshwater flux can be estimated (Serreze et al., 2006; Dickson et al., 2007; Bacon et al., 2015).

Considering the Arctic as a closed volume box, the surface freshwater flux ($F_{surf}$) can be approximated as:

$$F_{surf} \approx (\frac{\delta S}{\bar{S}}) V_o$$

where $\delta S$ is the difference in salinity between import and export from the box, $\bar{S}$ is the boundary mean salinity, and $V_o$ is the boundary volume flux (Bacon et al., 2015).

Until recently, budgets have been constructed using heterogeneous and asynoptic compendia of data which, through many years of work, are now beginning to tell a consistent story, though there is still uncertainty in all the major terms (e.g. Serreze et al., 2006; Dickson et al., 2007; Haine et al., 2015). Recent work by Tsubouchi et al. (2012, hereafter TB12), using quasi-synoptic ocean measurements from summer 2005, applied the commonly used box-inverse model technique (Wunsch, 1978) for estimating ocean volume fluxes to the Arctic. This approach represents a significant advance in Arctic freshwater flux estimates, resulting in the calculation of consistent optimised ocean velocity fields and the first quasi-synoptic estimates of Arctic freshwater and heat fluxes (TB12).

Salinity is used to quantify Arctic freshwater fluxes (Haine et al., 2015) because it responds only to dilution and concentration through the addition or removal of freshwater, respectively. Thus, it can be used to estimate unambiguously net changes in ocean freshwater fluxes. While marine measurements of salinity are routinely made to high precision,





enabling a precise estimate of the difference in salinity between import and export from the Arctic to be made ($\delta S$ above), accurate estimates of freshwater flux require the definition of an appropriate reference salinity ($\bar{S}$). Pragmatically, following Aagaard and Carmack (1989), this has been taken to be a notional Arctic mean salinity, though some investigators

have used different study-specific values (e.g. Dodd et al., 2012). A more recent theoretical treatment of the role of salinity has concluded that the boundary-mean salinity is the only appropriate reference salinity in the case of any (actual or notional) closed-volume freshwater budget (Bacon et al., 2015).

Salinity is not the only tracer that can be used to determine surface freshwater flux.
Ocean waters that have not been subject to significant evaporation/precipitation display a near-constant ratio of oxygen isotope concentrations (Craig, 1961) and, as water progresses round the hydrological cycle, isotopic fractionation (evaporation/freezing and precipitation/melting) alters this ratio ($\delta^{18}O$ Östlund and Hut, 1984). Hence, waters of meteoric origin (precipitation, river runoff), and those that have been ice-modified have distinct $\delta^{18}O$
values that can be used to decompose water samples into meteoric origin or ice-modified fractions (Östlund and Hut, 1984). The $\delta^{18}O$ tracer is conservative, reflecting only the net isotopic fractionation that the water sample has undergone. However, unlike salinity where freshwater has a definite salinity of zero, there is much variety in the $\delta^{18}O$ values observed for sea ice, river runoff (Bauch et al., 1995), and glacier ice (Cox et al., 2010). Following
Östlund and Hut (1984) there have been many studies using $\delta^{18}O$ to determine fractions of ice melt and meteoric water in the Arctic, most notably in the Fram Strait (Dodd et al., 2012; Meredith et al., 2001; Rabe et al., 2013), in the Canada Basin (Yamamoto-Kawai et al., 2008), and in the East Greenland Current (Cox et al., 2010). In terms of $\delta^{18}O$ signal, precipitation/evaporation and freezing/melting are manifestations of the same process with op-
posite signs. Consequently, $\delta^{18}O$ values reflecting only net isotopic fractionation are unable to quantify river runoff without the use of another conservative tracer. Initial work suggested the use of barium as a potential tracer of riverine input into the Arctic (Kenison Falkner et al., 1994). However, this tracer has recently been found to be non-conservative (through biological scavenging) in seawater (Abrahamsen et al., 2009).





Seawater in the North Pacific has a distinctly different biogeochemical composition from that in the North Atlantic, with Pacific seawater having higher concentrations of both the inorganic nutrients silicate and phosphate (Bauch et al., 1995; Ekwurzel et al., 2001; Jones et al., 1998). Silicate and nitrate concentrations (used in combination with oxygen; Ekwurzel et al., 2001), are only quasi-conservative as both are altered due to biological activity or air-sea exchange in surface waters (Alkire et al., 2015), while the use of nitrate:phosphate (N:P) nutrient ratios (Jones et al., 1998) is considered to be conservative with respect to biological activity. However, despite the N:P ratios for the Atlantic and Pacific exhibiting distinct linear relationships with near-constant slopes, there is variation in the exact form of this relationship (Jones et al., 2008; Sutherland et al., 2009; Dodd et al., 2012; Yamamoto-Kawai et al., 2008). In the Arctic Ocean, nutrient ratios have been used to trace the circulation of Pacific seawater (Jones et al., 1998; Jones, 2003), and to indicate the likely origins of freshwater sources (Yamamoto-Kawai et al., 2008; Sutherland et al., 2009).

Recent use of fixed ocean installations (moored current meters with temperature and salinity sensors) describing a closed circuit around the Arctic boundary by TB12 has enabled the first quasi-synoptic calculation of surface fluxes of heat and freshwater for the whole Arctic. We envisage that sustained measurement of suitable tracers around the Arctic boundary has the potential to generate estimates of surface freshwater fluxes with distinct meteoric and ice melt contributions. Such quantitative estimates would go a long way to mitigating the scarcity of Arctic observations, and represent a significant advance in Arctic science. Our aims in this study are twofold; (1) to combine two approaches for indirect estimations of freshwater flux using different and distinct tracers, and (2) to test the consistency of the various tracers used. To these ends, we aim to use existing nutrient and $\delta^{18}O$ data in combination with the salinity and optimised horizontal velocity fields of TB12 to estimate fluxes of meteoric, ice melt and oceanic source waters. We first describe the data sources and the model used along with the attribution method and schema implemented (Sect. 2). Results are presented (Sect. 3), and discussed with an examination of the implications for the future use of biogeochemical tracers in the Arctic (Sect. 4).





## 2 Data and methods

### 2.1 Measurements

TB12 use an inverse model (Wunsch, 1978; Roemmich, 1980) that considers the Arctic Ocean as a control volume bounded by land and four gateways – Davis, Fram and Bering Straits, and the Barents Sea Opening (Fig. 1) – and is divided into 15 horizontal layers defined by isopycnal surfaces. The TB12 inverse model generates an optimised horizontal velocity field $v(s,z)$, where $z$ is depth and $s$ the along-boundary horizontal coordinate, which conserves volume and salinity transports, based on hydrographic data collected in summer 2005. For further details of the inverse model construction see TB12. For this study, the TB12 volume fluxes are combined with additional tracers to generate source component estimates of liquid Arctic freshwater fluxes, to compare with the existing net (salinity-derived) estimates of TB12.

From the TB12 model, the Arctic boundary circulation is broadly conventional. Atlantic-origin seawater enters through the Barents Sea Opening with volume flux of $3.6 \pm 1.1$ Sv. Pacific-origin seawater enters through Bering Strait with volume flux of $1 \pm 0.2$ Sv. Fram Strait is a net exporter of seawater, with volume flux of $1.6 \pm 3.9$ Sv, representing a balance between inflowing (mainly) Atlantic waters in the West Spitsbergen Current in the east of the strait (volume flux of $3.8 \pm 1.3$ sV) and outflowing waters in the East Greenland Current in the west of the strait (volume flux of $5.4 \pm 2.1$ Sv). The net seawater export through Davis Strait has a volume flux of $3.1 \pm 0.7$ Sv. For details of other, relatively small contributions to the total, see TB12. As a simplified and approximate summary, $\sim 8$ Sv of Atlantic-origin and $\sim 1$ Sv of Pacific-origin seawater enters the Arctic, with $\sim 9$ Sv of variously modified seawater exported. The net surface freshwater flux calculated by TB12 is $187 \pm 44$ mSv, manifest as $147 \pm 42$ mSv in the liquid ocean plus $40 \pm 14$ mSv in sea ice.

Biogeochemical data were originally collated and published by Torres-Valdés et al. (2013) for inorganic nutrients and MacGilchrist et al. (2014) for $\delta^{18}O$. Original data sets are described as follows. For Davis Strait: Lee et al. (2004) (with additional data for 2005 supplied by Dr. Kumiko Azetsu-Scott, Department of Fisheries and Oceans, Bedford Insti-





tute of Oceanography). For Bering Strait: Woodgate et al. (2015). For Barents Sea Opening: The International Council for the Exploration of the Sea Oceanographic Database (http://ices.dk/ocean) for nutrient data, with Schmidt et al. (1999) for $\delta^{18}O$. For Fram Strait: Budéus et al. (2008); Kattner (2009) for nutrient data, with Rabe et al. (2009) for $\delta^{18}O$.
Sample locations are shown in Fig. 1. All nutrient and $\delta^{18}O$ data were optimally interpolated (Roemmich, 1983) in pressure both vertically and horizontally to match the station positions used by TB12 (Fig. 2).

Our domain comprises a total of 147 hydrographic stations (including 16 general circulation model grid cells in the Barents Sea Opening that function as hydrographic stations)
covering a total oceanic distance of 1803 km, with a total (vertical) section area of 1050 km$^2$. Vertical resolution is 1 dbar, with maximum pressures of 1044 dbar in Davis Strait, 2704 dbar in Fram Strait, 471 dbar in the Barents Sea Opening, and 52 dbar in Bering Strait (for further discussion of the model domain see TB12).

## 2.2 Approach

Following established practice, the freshwater content of a parcel of seawater is considered to have originated from a number of sources – typically three or four. The sources are characterised by end-members, which are defined points in the phase space populated by the observed seawater (and ice) biogeochemical (tracer) properties. Here we employ three variants of the approach to the calculation of the resulting source fractions. Firstly
a three end-member scheme (3EM) is adopted, which uses salinity and $\delta^{18}O$ to identify "plain" seawater, freshwater of meteoric origin, and ice-modified seawater. Secondly the 3EM scheme is extended to a four end-member scheme (4EM) through the use of inorganic nutrient data, aiming to discriminate between seawater of Atlantic and Pacific origin, where the salinity and $\delta^{18}O$ end-member properties of both ocean sources are assumed to be the
same as for Atlantic seawater. Thirdly the 4EM scheme is applied again, but now adopting distinct end-member properties for both ocean-source salinity and $\delta^{18}O$ (4EM+), replicating previous practice (Dodd et al., 2012; Jones et al., 2008; Sutherland et al., 2009).



To discriminate between freshwater of Atlantic and Pacific seawater origin, an additional relationship is formulated in terms of the concentrations of the inorganic nutrients phosphate and nitrate (Dodd et al., 2012; Jones et al., 1998). We form this relationship in terms of the variable $P^*$, which is an expression describing the excess concentration of phosphate above that which would be expected from typical Redfield nutrient ratios (Redfield et al., 1963), and it employs the observed nitrate concentration:

$$P^* = P_m - (N_m/16),$$

where $P_m$ and $N_m$ are the measured nitrate and phosphate concentrations, respectively. Atlantic and Pacific seawaters are each considered to have a distinct, near-constant, nitrate to phosphate (N:P) ratio (Jones et al., 1998), which can be expressed algebraically as:

$$P_{est} = P_{slope}N_m + P_{int},$$

where $P_{est}$ is the estimated concentrations of phosphate from the relevant ocean (either Atlantic and Pacific) and the subscripts $slope$ and $int$ indicate the slope and intercept of the relationships. Boundary sections of salinity, $\delta^{18}O$ and $P^*$ are shown in Fig. 2.

To quantify freshwater source fractions for each water parcel (i.e. grid point), we establish the following system of equations. This problem is conventionally treated as "square", with the number of constraints equal to the number of source water fractions to be determined for each water parcel. Each water parcel then has a suite of $i = 1, \ldots, M$ measured properties $x_i$. Each measured property is treated as the sum of $j = 1, \ldots, M$ fractions $f_i$ of a suite of source properties $X_{i,j}$. The number of source properties (or end-members) is here $M = 3$ or 4, and the associated freshwater sources are indicated as sea ice ($j = 1$), meteoric ($j = 2$), seawater ($j = 3$ for 3EM), or Pacific and Atlantic seawater ($j = 3$ and 4 for 4EM variants, respectively). Written as a sum:





$$X_i = \sum_{j=1}^{M} X_{i,j} f_j.$$

Setting all $x$, $X = 1$ for $i = 1$ retrieves the requirement that the sum of all the source fractions $f_j$ accounts for all of the observed seawater:

$$1 = \sum_{j=1}^{M} f_j. \tag{1}$$

The measured properties are then $\delta^{18}O$ concentrations ($i = 2$) and salinity ($i = 3$) for all models; in addition the 4EM variants employ $P^*$ for $i = 4$. The product of this process is a system of $M$ equations describing $M$ unknowns, which is written in matrix form for ($M \times 1$) column vectors $\mathbf{f}$ and $\mathbf{x}$, and ($M \times M$) matrix $\mathbf{X}$:

$$\mathbf{x} = \mathbf{X}\mathbf{f}.$$

This is solved for $\mathbf{f}$ by standard (exact) inversion of a square matrix at each water parcel on our ocean boundary grid, to calculate the resulting spatial distributions of the relevant freshwater source fractions:

$$\mathbf{f} = \mathbf{X}^{-1}\mathbf{x}.$$

## 2.3 End-member values

Previous studies have used different values for the end-member concentrations of salinity, $\delta^{18}O$ and nutrients, which are summarised in Tables 1 and 2. A least-squares linear fit to





the $\delta^{18}O$ and salinity data from the three sections likely to contain freshwater of meteoric origin (Davis, Fram and Bering Straits) suggests a $\delta^{18}O$ end-member in the range of -20 ‰ (Bering Strait) to -30 ‰ (Fram Strait), with a mean value of -23.3 ‰, which is within the range of the published values. The Fram Strait data exhibit the two-layer mixing relationship indicating the likely presence of Greenland ice sheet melt, which has a distinctly lighter $\delta^{18}O$ signature (Cox et al., 2010). The fits to data from the three sections likely to contain Atlantic seawater (Fram and Davis Straits, Barents Sea Opening) suggest an Atlantic seawater salinity endpoint of $\approx 35$. The relationships between salinity and $\delta^{18}O$ for our data and from cited sources are shown in Fig. 3A.

Considering the published nitrate-phosphate relationships, the most appropriate to this study are the values used by Jones et al. (2008), Sutherland et al. (2009), and Dodd et al. (2012), because Yamamoto-Kawai et al. (2008) include ammonium, and the nutrient measurements used here are of nitrate plus nitrite (Torres-Valdés et al., 2013). A least-squares best fit to the Bering Strait nutrient data has a slope of 0.0654, which is consistent with that of Jones et al. (2008), and an intercept of 0.6766. The relationships between nitrate and phosphate concentrations for our data and from cited sources are shown in Fig. 3B.

## 2.4  Freshwater flux calculation

We use the approach established by TB12 and developed by Bacon et al. (2015), which recognises that a unique definition of a freshwater flux is given by the net surface exchange between the ocean (including ice) and the adjacent land and atmosphere: i.e. the net of precipitation, evaporation and runoff. Then (using volume transports) the surface freshwater flux $F$ plus the ice and ocean boundary transport $V_O$ is conserved:

$$F + V_O = 0,$$

where





$$V_O = \oiint v(s,z)\, ds dz.$$

Lastly, the fraction of the ocean seawater flux per water parcel attributed to each of $n$ sources, $\delta V_O$ is:

$$\delta V_{O,j}(s,z) = f_i(s,z)v(s,z)\delta s \delta z,$$

⁵  where $\delta s$, $\delta z$ describe the horizontal and vertical grid spacing (or water parcel size).

## 2.5  End-Member uncertainty

Due to the wide range of plausible end-member values for each of the water types, to give an estimate of the likely uncertainty due to end-member choice, fluxes of the different water types were evaluated using a Monte-Carlo technique. Distributions for the different
¹⁰ end-member parameters were constructed from the cited values (Table 1) by assuming the parameter variability is normally distributed, with mean equal to the mean of the cited values and standard deviation equal to the range. A sample set of 1000 ensembles was drawn from the set of constructed parameter distributions using a Latin Hypercube sampling strategy (McKay et al., 1979). The distributions of the individual parameters in the ensemble, which
¹⁵ in all cases encompass the end points in Sect. 2.3 above, are shown in Fig. 4. Seawater salinity for 3EM and 4EM models is fixed at the boundary-mean salinity for the TB12 model (34.662).

For each model approach, fluxes of the different water types were estimated by combining the velocities from the TB12 model with the calculated water type fractions for the sample
²⁰ ensemble. Mean and standard deviations for the attributed volume fluxes of each water type were calculated as the mean and standard deviation of the results from the sample ensemble.



## 3 Results

Here we present the results of the application of the methods and end-members, described in Sect. 2, to generate three and four end-member freshwater source fractions and fluxes. Equation 1 allows for individual fractions to be either $< 0$ or $> 1$ as long as the sum of
all fractions is equal to one. There is a valid physical interpretation for negative fractions of meteoric and ice-modified waters, where processes remove freshwater from seawater representing net evaporation and sea ice formation, respectively. Seawater fractions, either total or individual Atlantic and Pacific water fractions, should be positive. Consequently, Pacific and Atlantic water fractions were made positive-definite by rounding to zero any of
the fractions that were less than zero, and setting the remaining seawater fraction so that equation 1 was not invalidated.

### 3.1   Three end-member model (3EM)

The distribution of 3EM source fractions is shown in Fig. 5. Ice-modified waters are found almost exclusively in the surface / upper waters of the model (depths down to 1000 dbar
in the Davis Strait), with highest-magnitude fractions ($-0.15$) found in sub-surface waters of the western Fram Strait between depths of $\sim 50$ and 300 dbar. The fractions of ice-modified waters are mostly negative (indicating high salinity from upstream ice formation), with a small fraction ($\sim 0.05$) positive (indicating fresh meltwater) in the surface (above 70 dbar) East Greenland Current (EGC; between 6.5 and 2° W) of the Fram Strait. Meteoric
waters are also found almost exclusively in the surface / upper waters of the model, with high fractions ($> 0.08$) in the surface / sub-surface waters (depths down to 350 dbar) in the Davis Strait and the western side of the Fram Strait. There is also a high fraction of meteoric water in the Bering Strait. Seawater fractions are high ($\sim 1$) in all deep / intermediate model waters at depths in excess of $\sim 350$ dbar.
Typical volume fluxes for the 3EM source fractions are shown in Fig. 6. The strongest fluxes of ice-modified waters occur in surface waters of the middle of the Davis Strait, on the western side of the Fram Strait (EGC) and to the east in the Bering Strait, with fluxes



of $\sim 0.1$ Sv in magnitude (positive fluxes indicating an export of high-salinity waters). The patterns of countervailing fluxes over the Belgica Bank (west of $6.5°$ W) in the Fram Strait are indicative of recirculation (see TB12). Meteoric water volume fluxes follow the same general pattern as for ice-modified waters, with strong export ($\sim 0.1$ Sv) in the middle of the Davis Strait and the EGC. There is a strong import ($\sim 0.1$ Sv) of meteoric waters in the Bering Strait. Seawater volume fluxes indicate a strong export mid-Davis Strait ($\sim 1$ Sv), with moderate export in the EGC ($\sim 0.5$ Sv) and moderate import to the east in the Fram Strait in the West Spitsbergen Current (WSC; east of $5°$ E) and in the Bering Strait. There is also weak export of seawater ($\sim 0.1$ Sv) for the deeper waters in the middle of the Fram Strait (between $2°$ W and $5°$ E).

For the 3EM model schema, the net seawater volume flux is effectively zero ($0.002 \pm 0.006$ Sv, Table 3). The net volume export of meteoric waters ($200 \pm 44$ mSv) is consistent with the TB12 surface freshwater input of $187 \pm 44$ mSv (Table 3). The model also indicates a volume of exported high-salinity ice-modified water ($60 \pm 50$ mSv), which is consistent with the model solid sea ice export of $40 \pm 14$ mSv, with the bulk of this export occurring through the Davis Strait (note that this is represented in the model by an apparent net import of fresh ice-modified water in opposition to the general circulation, Table 3).

The 3EM model indicates that the volume export of meteoric water through Fram Strait is concentrated in the Belgica Bank and EGC regions ($22 \pm 6$ (mSv) and $83 \pm 50$ (mSv), respectively), with close to zero meteoric flux in the remainder of the strait (Table 4). This is consistent with the picture described in previous studies Dodd et al. (2012); Rabe et al. (2009); Meredith et al. (2001). High-salinity ice-modified water is exported mainly in the EGC ($88 \pm 56$ mSv), with small ($\sim 5$ mSv) fluxes of ice-modified water in the middle and Belgica Bank sections of the strait (Table 4). The import of high-salinity water in the WSC is attributed exclusively to ice-modified water ($44 \pm 36$ mSv), reflecting the higher $\delta^{18}O$ values at the surface ($\sim 0.4$ ‰) relative to those for deeper waters ($\sim 0.2$ ‰)) to the east of $5°$ W (Fig. 2).





## 3.2 Four end-member models (4EM and 4EM+)

The distributions of 4EM and 4EM+ source fractions are shown in Figures 7 and 9, respectively, and characteristic volume fluxes for the source fractions in Figures 8 and 10. In common with the 3EM model, both 4EM and 4EM+ models allocate the bulk of the ice-modified water to the surface / upper waters. However, both four end-member schema indicate a non-zero fraction ($\sim 0.01$) of ice-modified waters in the deeper waters of the Fram Strait and Barents Sea Opening. The distribution of meteoric waters in both four end-member models is consistent with the 3EM model where meteoric waters also mostly occupy the surface layers. However, differences occur in the Davis Strait, where the 4EM and 4EM+ models indicate lower fractions ($\sim 0.01$) below $\sim 350$ dbar, in the Bering Strait where meteoric water is confined to the eastern side, and in the deeper waters of the model where the meteoric fraction is non-zero ($< 0.01$). Both four end-member models indicate Pacific water mostly in the surface / near surface waters of the Davis, Fram and Bering Straits, and almost exclusively Atlantic Water in the deepest waters of the model ($\sim 0.9$). Both models indicate quantities of Pacific Water in the deepest water of the Fram Strait and Barents Sea Opening ($\sim 0.1$), and Atlantic water in the Bering Strait ($\sim 0.1$).

Differences between the 3EM and four end-member model schema are also reflected in the fluxes of the different fractions. For both four end-member models, there are non-zero fluxes of ice-modified waters, meteoric water (both $< 0.005$ Sv), and Pacific water ($< 0.02$ Sv) in the deeper waters of the Fram Strait and Barents Sea Opening. Consistent with the 3EM model, the 4EM model has a net oceanic volume flux (sum of Pacific and Atlantic contributions) that is effectively zero (4EM $0.002 \pm 0.006$ Sv, Table 5), while the net oceanic volume flux for the 4EM+ model is larger and indicates a net outflow ($-0.104 \pm 0.051$ Sv, Table 7). Net model liquid freshwater export (sum of meteoric and ice-modified fractions) for the 4EM model is the same as for the 3EM model ($140 \pm 67$ mSv), while the 4EM+ export is smaller with a large uncertainty ($35 \pm 51$ mSv).

Net ice-modified water flux for both the 4EM and 4EM+ schemas is also consistent with the 3EM model and the TB12 solid ice flux, with the 4EM model estimating $60 \pm 50$ mSv




and the 4EM+ $63 \pm 64$ mSv (Tables 5 and 7). Both 4EM and 4EM+ models show the same flux pattern for ice-modified water as the 3EM model, with the bulk of the high-salinity ice-modified water exiting through the Davis Strait (Tables 3, 5 and 7, Fig. 11).

While the net volume flux of meteoric water for the 4EM model is the same as that of the 3EM ($200 \pm 44$ mSv), the 4EM+ model estimates a smaller net volume flux ($98 \pm 46$ mSv, Tables 5 and 7). Both 4EM and 4EM+ models show the same flux pattern for meteoric water as the 3EM model, with meteoric water entering the Bering Strait and exiting through the Davis and Fram Straits. However, the net import of meteoric water through the Bering Strait and the net export of meteoric water through the Davis Strait in the 4EM+ model schema is approximately half the magnitude of the fluxes in the other two schema (Tables 3, 5 and 7, Fig. 11).

Both 4EM and 4EM+ model schemas indicate an imbalance in the net volume fluxes for both Pacific and Atlantic seawater, with both model schemas showing a net export of Pacific seawater (4EM $1.495 \pm 0.268$ Sv; 4EM+ $1.488 \pm 0.263$ Sv) that is balanced by a net import of Atlantic seawater of approximately equal magnitude (4EM $1.497 \pm 0.268$ Sv; 4EM+ $1.384 \pm 0.255$ Sv, Tables 5 and 7). Current understanding of Arctic fluxes suggests that Pacific water enters the Bering Strait and exits both through the Davis Strait, after passing through the western Canadian Archipelago, and on the western side of the Fram Strait (Haine et al., 2015). Consistent with this view, both four end-member schemas indicate that Pacific water, entering the Arctic through the Bering Strait, exits mostly through the Davis Strait with a much ($O(10\times)$) smaller flux through the Fram Strait, mainly across Belgica Bank and in the EGC. Export of Pacific water through the Davis Strait is approximately twice the magnitude of the import through the Bering Strait (Tables 5 and 7). Atlantic seawater circulates in through the Barents Sea Opening and out through the Fram and Davis Strait, with the import through the Barents Sea approximately twice the magnitude of the export (Tables 5 and 7).

For the Fram Strait, the pattern of water fluxes described by both the 4EM and 4EM+ schemas is consistent with the pattern described above for the 3EM model (Tables 6 and 8). In both four end-member schema, Pacific-origin water is exported across Belgica Bank



and in the EGC, accounting for approximately 15% of the Fram Strait oceanic volume flux (Tables 6 and 8). While fluxes of meteoric and ice-modified waters described by the 4EM model are the same as for the 3EM model (Table 6), the fluxes from the 4EM+ schema are different (Table 8).

5    The description of Arctic freshwater fluxes presented by the 4EM+ model is broadly consistent with that from previous studies of fluxes in the Fram Strait using 4EM+ type schemas with distinct Pacific seawater, $\delta^{18}O$, and salinity end-members (Dodd et al., 2012; Azetsu-Scott et al., 2012; Rabe et al., 2013). Analysis of a time series of observations from the Fram Strait suggest a mean freshwater export flux dominated by waters of meteoric origin, mixed with high-salinity ice-modified waters to the west of $2°$ W in the EGC and over the Greenland shelf (i.e. Belgica Bank), with fluxes of negative meteoric origin waters also noted in the WSC (Dodd et al., 2012; Rabe et al., 2013).

The greatest differences between the models are in the fluxes of meteoric and ice melt waters across Belgica Bank and in the EGC (Fig. 11), with the 4EM+ schema showing less export of meteoric water in the EGC compared to the the other schema. In the 4EM+ model, the import of high-salinity water in the WSC is attributed almost equally to "negative" meteoric-origin water and high-salinity ice-modified water, in contrast to the 4EM and 3EM schema, which attribute this high-salinity import to high-salinity ice-modified water (Tables 6 and 8). Export of ice-modified water is also lower in the 4EM+ schema compared to the 3EM and 4EM models (Tables 6 and 8; Fig. 11).

The estimated distribution of water types across the Davis Strait in the 4EM+ model is qualitatively consistent with previous studies, where column inventories of the water types show highest freshwater content on the western side of the strait, where the net freshwater inventory consists of a mixture of "oceanic freshwater" and high-salinity ice-modified water (Azetsu-Scott et al., 2012). To the east of the Davis Strait, there is a contribution from fresh ice-modified water (Azetsu-Scott et al., 2012).





## 4   Discussion and summary

In this section, we first examine points of consistency, both between the different end-member models and between the models and other evidence; then we consider inconsistencies, and their consequent meaning, between those models; finally, we offer some general perspectives on freshwater calculations in the Arctic.

### 4.1   Consistency

Within uncertainty, the net seawater flux of the 3EM and 4EM models is zero: $2 \pm 6$ mSv for 3EM; $2 \pm 379$ mSv for 4EM (Tables 3 and 5). Furthermore, the 3EM and 4EM model estimates of net Arctic meteoric freshwater volume export flux is $200 \pm 44$ mSv (Tables 3 and 5), which agrees well (again, within uncertainty) with the TB12 surface freshwater input of $187 \pm 44$ mSv.

Sea ice is frozen out of liquid seawater, leaving behind in the seawater a negative $\delta^{18}O$ signal resulting from this distillation-type process (Östlund and Hut, 1984). In the long-term mean, and allowing for trends in net freshwater input and lags between this input at the surface and its manifestation at the boundary, the positive freshwater export flux of the sea ice should be approximately equal to the negative freshwater export flux of the freshwater deficit resulting from this sea ice formation. We find the latter (the deficit flux) to be $60 \pm 50$ mSv for both the 3EM and 4EM models, and this is similar to the TB12 sea ice export of $40 \pm 14$ mSv. The TB12 measurements were made in summertime, and we note evidence of seasonal signal "cancellation", where the $\delta^{18}O$ seawater deficit signal is a maximum at depth ($\sim 50$ m; Figures 5 and 7) and reduces towards the surface, which we interpret as the (seasonal) result of sea ice melting back into the near-surface seawater.

The previous two paragraphs note that (i) the TB12 net surface freshwater flux is (approximately) the same as our net meteoric freshwater flux, and (ii) our sea ice and ice-modified water fluxes are (approximately) equal and opposite. A further, combined, view arises. The net 3EM and 4EM liquid freshwater export is the sum of the meteoric and ice-modified freshwater fractions, and equals $140 \pm 67$ mSv, which is the same as the TB12 net liquid





freshwater export of $147 \pm 42$ mSv (the total freshwater flux is then obtained by adding the solid, sea ice, fraction). Fluxes of liquid freshwater from the TB12 model also compare well to the 3EM and 4EM models' net surface volume fluxes of meteoric and ice-modified waters across the four main gateways (Fig. 12).

## 4.2 Inconsistency

The first inconsistency arises from the inclusion of inorganic nutrient constraints. In the 4EM model, $\sim 1$ Sv of Pacific seawater enters the Arctic through Bering Strait, while $\sim 2.5$ Sv of Pacific seawater exits the Arctic, mainly through Davis Strait, indicating the apparent net "creation" of $\sim 1.5$ Sv of Pacific seawater (Table 5). The quantity of seawater labelled "Pacific" that exits the Arctic (mainly through Davis Strait) is more than double the quantity of actual Pacific seawater entering (through Bering Strait). This is mirrored by the origins and fate of Atlantic seawater, with $\sim 3.6$ Sv entering the Arctic and only $\sim 2.1$ Sv exiting, which indicates an apparent net "destruction" of $\sim 1.5$ Sv of Atlantic seawater (Table 5). The magnitude of this apparent "conversion" of Atlantic to Pacific seawater is over five times greater than the uncertainty on the fluxes ($\sim 0.3$ Sv; Table 5). In the 4EM model, discrimination between Atlantic and Pacific waters is solely based on $P^*$, which consequently suggests that the assumption that the nitrate:phosphate (N:P) nutrient ratio is a conservative tracer of seawater origins is flawed.

The N:P ratio (expressed here as $P^*$) was proposed as a tracer that would be conservative with respect to biological activity (Jones et al., 1998, 2008; Yamamoto-Kawai et al., 2008). However, evidence indicates that the N:P of waters entering the Arctic is further modified along their pathways, most likely by denitrification alone or in combination with a potential external source of phosphate (Torres-Valdés et al., 2013; Devol et al., 1997). This is particularly true for waters of Pacific origin, known to undergo further denitrification over the Chukchi Shelf, such that the $P^*$ signal in Davis Strait is much larger than at the Bering Strait (Torres-Valdés et al., 2013). Those modification processes thus render the N:P ineffective as a tracer (at least when considering full depth assessments).



Additionally, the N:P nutrient ratio of river runoff is pragmatically assumed to be constant and to match that of Atlantic seawater, in that it has no excess of phosphate (Dodd et al., 2012; Yamamoto-Kawai et al., 2008; Jones et al., 2008). However, knowledge of the riverine delivery of water constituents such as nutrients, sediment, and carbon is less well constrained than estimates of freshwater volume (Bring et al., 2016, 2017), suggesting that there may be other, as yet unquantified, riverine nutrient sources (or sinks) in the Arctic. Denitrification in the bottom sediments of the Laptev Sea continental margin, inferred from nutrient budget estimates over the shelf, has also been suggested to lead to a potential overestimate in the volume of Pacific origin seawater when using nutrient ratios as a tracer (Bauch et al., 2011). Although these factors are already acknowledged as likely sources of error when using N:P ratios as a tracer (Dodd et al., 2012; Yamamoto-Kawai et al., 2008; Jones et al., 2008), our results suggest that the influence of such processes is likely to be significantly greater than previously thought.

The second inconsistency arises from consideration of the composition and "labelling" of the waters of Bering Strait. Water entering the Arctic through the Bering Strait should, by definition, be seawater of Pacific origin. However, the Bering Strait inflow is unusually fresh because it contains a significant fraction of meteoric freshwater (Östlund and Hut, 1984, and Table 3). The meteoric water in the Bering Strait originates in part from the Alaskan Coastal Current on the east side of Bering Strait, and it preserves the runoff signal from the western North American rivers: (e.g. Woodgate and Aagaard, 2005; Chan et al., 2011). A second important reason for the presence of meteoric freshwater in Bering Strait is the basic fact that the Pacific Ocean experiences a net positive precipitation anomaly: (e.g. Warren, 1983). There are two sets of constraints on the water in Bering Strait, therefore: it must be all Pacific water (defined by $P^*$), because that is where it comes from; and it must be $\sim 10\%$ meteoric freshwater (defined by $\delta^{18}O$) to generate its low salinity. These constraints must, therefore, be partially degenerate (Fig. 3).

A third inconsistency arises when Pacific and Atlantic seawaters are defined as separate categories using salinity and $\delta^{18}O$. These two seawaters will lie on the mixing line between any single seawater category, such as that associated with the TB12 boundary-





mean salinity, and pure freshwater (Fig. 3). If Pacific seawater lies on this mixing line and is also defined as a separate category, then these constraints are also degenerate.

The results of using such wholly or partially degenerate constraints on the model fluxes are most clearly manifested in the 4EM+ model. In contrast to the models with common seawater properties (3EM and 4EM), there is a positive net ocean volume export for the 4EM+ model ($104 \pm 51$ mSv). The fraction of Pacific origin seawater identified in the 4EM+ model is not significantly different from that in the 4EM model ($0.806 \pm 0.076$ Sv, for 4EM; $0.825 \pm 0.099$ Sv, for 4EM+). However, the volume of meteoric water identified is about half that of the 4EM model (Tables 5 and 7). The TB12 net salinity-based estimate of liquid freshwater export compares well to the $\delta^{18}O$-derived estimates of meteoric origin waters and high-salinity ice-modified water from the 3EM and 4EM models, which is consistent with the current paradigm of the Arctic freshwater budget (Haine et al., 2015). However, for the 4EM+ model the picture of Arctic freshwater export being the sum of meteoric and ice-modified water fractions is modified by the inclusion of an "oceanic" origin freshwater component (Fig. 12). The theoretical underpinning of the definition of a single reference salinity used in the 3EM and 4EM models (Bacon et al., 2015) combined with the constancy of $\delta^{18}O$ in oceanic waters, which is the basis of the use of $\delta^{18}O$ as a tracer (Östlund and Hut, 1984), leads us to the interpretation of the 4EM+ model Pacific water as a mixture of seawater, meteoric water and ice-modified water in an undefined ratio. Consequently "oceanic" origin freshwater, in the 4EM+ schema, is likely to be simply a mixture of meteoric water and ice-modified water in undefined ratio. This interpretation of oceanic origin freshwater is consistent with the results for the Fram and Davis Straits, where an increase in oceanic freshwater flux is matched by a decrease in predicted meteoric freshwater flux (Figures 11 and 13).

## 4.3 Perspectives

Continuing the point of discussion from the previous section: the use of the N:P inorganic nutrient ratio as a tracer can appropriately distinguish Atlantic and Pacific origin seawater on entry to the Arctic (Jones et al., 1998, and Fig. 3).




Our evidence indicates an apparent ∼ 1.5 Sv conversion of inflowing Atlantic water into a water mass ("Polar", perhaps) with the same inorganic nutrient properties as inflowing Pacific water. Geochemically, processes which change the N:P ratio are observed to occur in the shallower shelf regions of the Arctic and in the Chukchi Sea (Devol et al., 1997; Chang and Devol, 2009; Bauch et al., 2011). Consequently, this conversion may have been achieved through further modification of inflowing Pacific water which then mixes with Atlantic water prior to outflow. Alternatively, there may be a process (or processes) that we do not understand modifying Atlantic water directly within the Arctic. While the understanding of nutrient sources, sinks, and transformations within the Arctic remains incomplete (e.g. Torres-Valdés et al., 2013, 2016), we cannot ascertain exactly where or how the addition of phosphate and/or the removal of nitrate may be happening. However, we must now regard the employment of the N:P ratio in this context to be unsafe.

Turning now to positive results, this is the first demonstration of consistency between the "control volume" approach to quantification of freshwater fluxes (as in TB12) and the geochemical tracer approach, so we describe now how and why this works. The TB12 approach is outlined above in Sect. 2.1, is mathematically generalised in Bacon et al. (2015), and is further illustrated in (Carmack et al., 2016, Appendix). Traditional ocean (and sea ice) freshwater flux calculations have in the past required the use of arbitrary reference salinities. However, in this approach, there is nothing to distinguish freshwater from the pure water component of seawater (cf. Wijffels et al., 1992; Talley, 2008). The key perception that enabled the analytical removal of arbitrary reference salinities is that there is only one unique physical (and non-geochemical) definition of freshwater in the marine context: the net freshwater flux at the surface (meaning the net of precipitation, evaporation and runoff). For this approach to work, an actual (or notional) control volume, plus knowledge around the marine boundary of velocity and salinity, is required. The outcome is that the reference salinity in the freshwater flux calculation is functionally replaced by the ocean (and sea ice) boundary-mean salinity.

The approach here employs three valid and geochemically distinct categories of water: sea ice (in its various manifestations), meteoric (surface-origin) freshwater, and seawater





(where seawater is the component of the mixture that contains all of the dissolved salt and this contains no significant isotopic distillation signature). First, we note again that our total sea ice flux, being the sum of the fluxes of solid sea ice, sea ice meltwater, and the freshwater deficit in the seawater from which the ice was formed, is approximately zero. Second, the TB12 velocity field is constrained to conserve salinity, and this is reflected in our zero net seawater fluxes, which is another statement of salinity conservation, because "seawater" is the category that contains all of the ocean salinity. Third, we note that the same categories (both here and in TB12) of surface-origin freshwater are all meteoric. This is why our surface (meteoric) freshwater flux agrees with the TB12 results: both are (explicitly or implicitly) meteoric.

In conclusion, in this work we have both reconciled the (traditionally divergent) perspectives of the Arctic freshwater budget provided by control volume and geochemical approaches, and shed light into the causes of their previously conflicting results. Our findings indicate that future applications of geochemical approaches to monitoring the climatic evolution of Arctic freshwater fluxes should avoid tracer-based definitions of distinct oceanic water types.

## Data availability

All data used in the analysis presented here is available from the original authors. See Sect. 2.1 for details.

## Author contribution

AF conducted the analysis and prepared the manuscript. SB and ACNG assisted with the analysis and preparation of the manuscript. TT assembled the data used and assisted with manuscript preparation. STV assisted with manuscript preparation.

*Acknowledgements.* This study was funded by the U.K. Natural Environment Research Council as a contribution to the TEA-COSI (The Environment of the Arctic Climate, Ocean and Sea Ice) project



grant no. NE/I028947/1. ACNG acknowledges the support of the Royal Society and the Wolfson Foundation.



|  | **Atlantic** | **Pacific** | **Met.** | **Ice Melt** | **Source** |
|---|---|---|---|---|---|
| $\delta^{18}O$ (‰) | $0.24 \pm 0.03$ | $-0.8 \pm 0.1$ | $-20 \pm 2$ | $-2 \pm 1.0$ | Yamamoto-Kawai et al. (2008) |
|  | $0.3$ | $-1.0 \pm 0.5$ | $-21$ | $surf+2.1$ | Bauch et al. (1995) |
|  | $0.3$ | $-1.3$ | $-18.4$ | $0.5$ | Dodd et al. (2012) |
|  | $0.19 \pm 0.06$ | $-0.8 \pm 0.1$ | $-18 \pm 2$ | $-2 \pm 1$ | Azetsu-Scott et al. (2012) |
|  | $0.35 \pm 0.15$ | $-1 \pm 0.1$ | $-21 \pm 2$ | $1 \pm 0.5$ | Sutherland et al. (2009) |
| Sal. (PSU) | $34.87 \pm 0.03$ | $32.5 \pm 0.2$ | $0$ | $4 \pm 1$ | Yamamoto-Kawai et al. (2008) |
|  | $34.92$ | $33$ | $0$ | $3$ | Bauch et al. (1995) |
|  | $34.9$ | $32.0$ | $0$ | $4$ | Dodd et al. (2012) |
|  | $34.75 \pm 0.14$ | $32.5 \pm 0.2$ | $0$ | $4 \pm 1$ | Azetsu-Scott et al. (2012) |
|  | $35 \pm 0.15$ | $32.7 \pm 1$ | $0$ | $4 \pm 1$ | Sutherland et al. (2009) |

**Table 1.** End-member values for salinity and $\delta^{18}O$ (‰) from the literature.



|  | Slope | Intercept | Source |
|---|---|---|---|
| Atlantic | 0.0545 | 0.1915 | Jones et al. (2008) |
|  | 0.053 | 0.170 | Dodd et al. (2012) |
|  | $0.048 \pm 0.003$ | $0.130 \pm 0.04$ | Sutherland et al. (2009) |
| Pacific | 0.0653 | 0.94 | Jones et al. (2008) |
|  | $0.08 \pm 0.015$ | $0.85 \pm 0.13$ | Sutherland et al. (2009) |
|  | 0.0654 | 0.6766 | Calculated for this study from observations |

**Table 2.** P:N relationships, where $PO_4 = Slope * NO_3 + Intercept$ ($\mu$ mol kg$^{-1}$)




|  | **Oceanic** | **Met.** | **Ice Melt** | **Sum** |
|---|---|---|---|---|
| Davis | -3.035 ± 0.008 | -0.209 ± 0.055 | 0.100 ± 0.062 | -3.144 |
| Fram | -1.566 ± 0.004 | -0.104 ± 0.027 | 0.038 ± 0.030 | -1.632 |
| Barents | 3.671 ± 0.004 | 0.013 ± 0.031 | -0.048 ± 0.035 | 3.636 |
| Bering | 0.931 ± 0.003 | 0.099 ± 0.023 | -0.029 ± 0.026 | 1.001 |
| **Liquid** | 0.002 ± 0.006 | -0.200 ± 0.044 | 0.060 ± 0.050 | -0.139 |
| **Solid** |  |  | -0.040 ± 0.014 | -0.04 |

**Table 3.** Mean volume fluxes (Sv ± standard deviation) for the three end-member (3EM) model. Positive values indicate fluxes into the Arctic.



|  | Oceanic | Met. | Ice Melt | Sum |
|---|---|---|---|---|
| BB | -0.350 ± 0.001 | -0.022 ± 0.006 | -0.002 ± 0.006 | -0.373 |
| EGC | -5.364 ± 0.007 | -0.083 ± 0.050 | 0.088 ± 0.056 | -5.359 |
| Mid. | 0.303 ± 0.000 | -0.000 ± 0.003 | -0.005 ± 0.003 | 0.298 |
| WSC | 3.845 ± 0.004 | 0.001 ± 0.032 | -0.044 ± 0.036 | 3.803 |
| **Liquid** | -1.566 ± 0.004 | -0.104 ± 0.027 | 0.038 ± 0.030 | -1.632 |

**Table 4.** Mean volume fluxes (Sv ± standard deviation) for the components of the Fram Strait flux (Belgica Bank, BB; East Greenland Current, EGC; Mid-strait, Mid.; West Spitsbergen Current, WSC) from the three end-member (3EM) model. Positive values indicate fluxes into the Arctic.





|  | Atlantic | Pacific | Met. | Ice Melt | Sum |
|---|---|---|---|---|---|
| Davis | -0.815 ± 0.346 | -2.219 ± 0.346 | -0.209 ± 0.055 | 0.100 ± 0.062 | -3.144 |
| Fram | -1.333 ± 0.088 | -0.233 ± 0.088 | -0.104 ± 0.027 | 0.038 ± 0.030 | -1.632 |
| Barents | 3.520 ± 0.184 | 0.151 ± 0.184 | 0.013 ± 0.031 | -0.048 ± 0.035 | 3.636 |
| Bering | 0.126 ± 0.076 | 0.806 ± 0.076 | 0.099 ± 0.023 | -0.029 ± 0.026 | 1.001 |
| **Liquid** | 1.497 ± 0.268 | -1.495 ± 0.268 | -0.200 ± 0.044 | 0.060 ± 0.050 | -0.139 |
| **Solid** |  |  |  | -0.040 ± 0.014 | -0.04 |

**Table 5.** Mean volume fluxes (Sv ± standard deviation) for the four end-member (4EM) model. Positive values indicate fluxes into the Arctic.





|  | **Atlantic** | **Pacific** | **Met.** | **Ice Melt** | **Sum** |
|---|---|---|---|---|---|
| BB | -0.182 ± 0.035 | -0.167 ± 0.035 | -0.022 ± 0.006 | -0.002 ± 0.006 | -0.373 |
| EGC | -4.948 ± 0.376 | -0.416 ± 0.377 | -0.083 ± 0.050 | 0.088 ± 0.056 | -5.359 |
| Mid. | 0.226 ± 0.058 | 0.077 ± 0.058 | -0.000 ± 0.003 | -0.005 ± 0.003 | 0.298 |
| WSC | 3.571 ± 0.274 | 0.274 ± 0.275 | 0.001 ± 0.032 | -0.044 ± 0.036 | 3.803 |
| **Liquid** | -1.333 ± 0.088 | -0.233 ± 0.088 | -0.104 ± 0.027 | 0.038 ± 0.030 | -1.632 |

**Table 6.** Mean volume fluxes (Sv ± standard deviation) for the components of the Fram Strait flux (Belgica Bank, BB; East Greenland Current, EGC; Mid-strait, Mid.; West Spitsbergen Current, WSC) from the four end-member (4EM) model. Positive values indicate fluxes into the Arctic.





|  | Atlantic | Pacific | Met. | Ice Melt | Sum |
|---|---|---|---|---|---|
| Davis | -0.934 ± 0.343 | -2.231 ± 0.367 | -0.060 ± 0.057 | 0.080 ± 0.084 | -3.144 |
| Fram | -1.333 ± 0.079 | -0.234 ± 0.086 | -0.091 ± 0.025 | 0.026 ± 0.030 | -1.632 |
| Barents | 3.493 ± 0.168 | 0.151 ± 0.185 | 0.011 ± 0.037 | -0.019 ± 0.050 | 3.636 |
| Bering | 0.158 ± 0.089 | 0.825 ± 0.099 | 0.041 ± 0.030 | -0.023 ± 0.034 | 1.001 |
| **Liquid** | 1.384 ± 0.255 | -1.488 ± 0.263 | -0.098 ± 0.046 | 0.063 ± 0.064 | -0.139 |
| **Solid** |  |  |  | -0.040 ± 0.014 | -0.04 |

**Table 7.** Mean volume fluxes (Sv ± standard deviation) for the reference (4EM+) model. Positive values indicate fluxes into the Arctic.



| | Atlantic | Pacific | Met. | Ice Melt | Sum |
|---|---|---|---|---|---|
| BB | -0.191 ± 0.034 | -0.167 ± 0.035 | -0.011 ± 0.005 | -0.004 ± 0.007 | -0.373 |
| EGC | -4.929 ± 0.345 | -0.416 ± 0.376 | -0.060 ± 0.057 | 0.046 ± 0.073 | -5.359 |
| Mid. | 0.231 ± 0.053 | 0.076 ± 0.056 | -0.007 ± 0.004 | -0.003 ± 0.005 | 0.298 |
| WSC | 3.556 ± 0.251 | 0.274 ± 0.273 | -0.013 ± 0.040 | -0.014 ± 0.051 | 3.803 |
| **Liquid** | -1.333 ± 0.079 | -0.234 ± 0.086 | -0.091 ± 0.025 | 0.026 ± 0.030 | -1.632 |

**Table 8.** Mean volume fluxes (Sv ± standard deviation) for the components of the Fram Strait flux (Belgica Bank, BB; East Greenland Current, EGC; Mid-strait, Mid.; West Spitsbergen Current, WSC) from the four end member (4EM+) model. Positive values indicate fluxes into the Arctic.



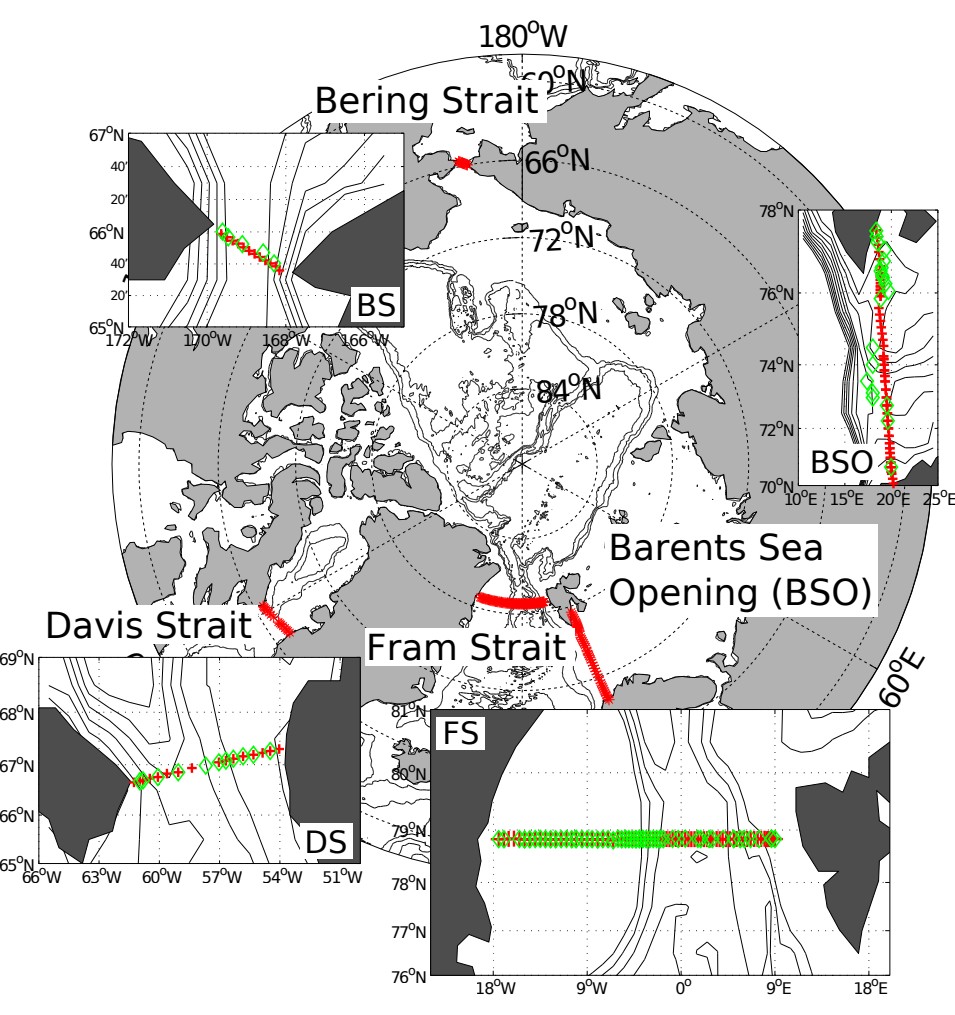

**Figure 1.** Map of the Arctic Ocean, showing the four main gateways. The position of the $\delta^{18}O$ and nutrient sample locations is indicated by green diamonds, and the Tsubouchi et al. (2012) CTD station positions by red crosses.



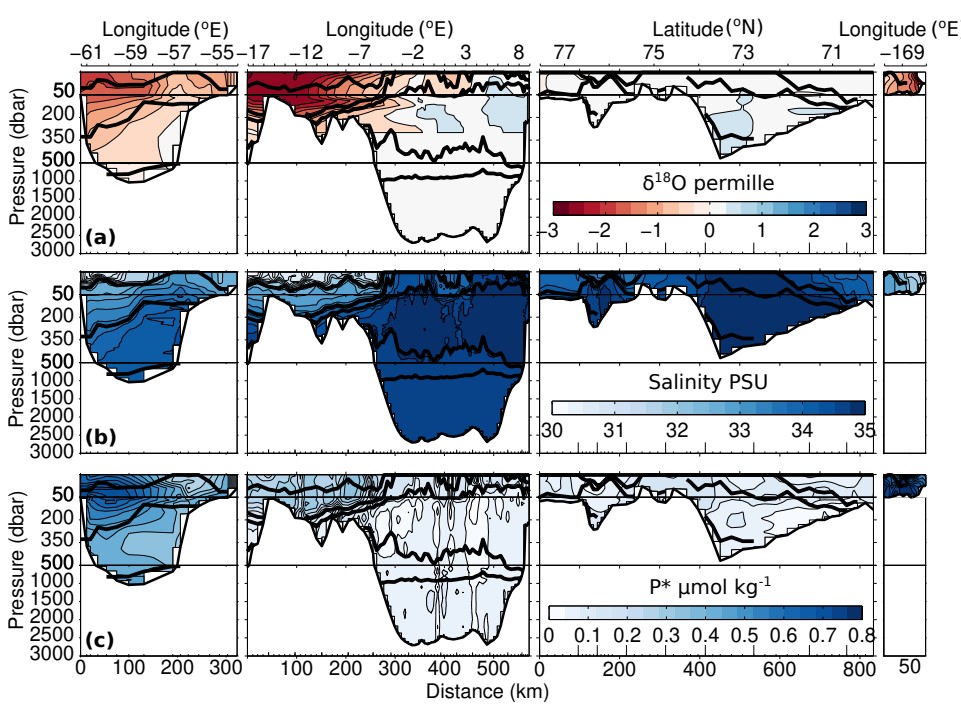

**Figure 2.** Sections of $\delta^{18}O$ (panel a) salinity (panel b) and $P*$ (panel c), after optimal interpolation onto the Tsubouchi et al. (2012) CTD station positions, clockwise around the four gateways from Davis to Bering Straits. Solid black lines indicate the isopycnal surfaces separating the main Arctic water masses, as described in Tsubouchi et al. (2012).





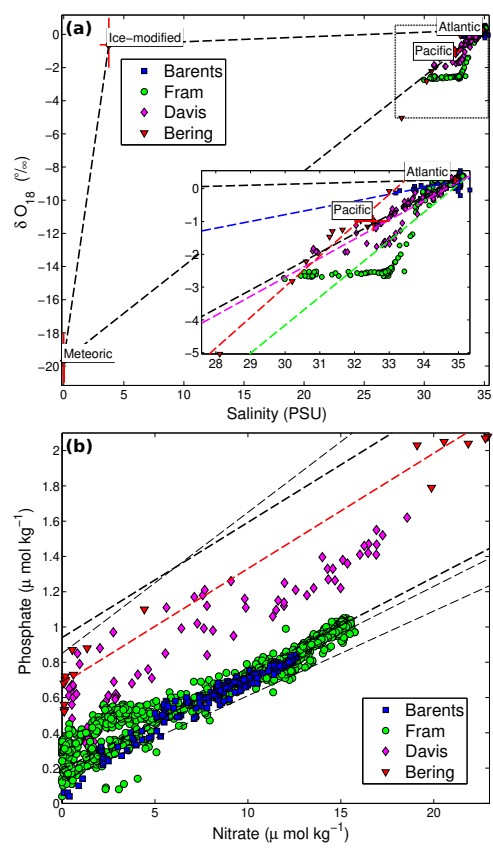

**Figure 3.** Panel a: Salinity - $\delta^{18}O$ relationship for all samples used in this manuscript; mean literature end-points ($\pm$ standard deviation) are marked. The dashed lines in the inset indicate a linear best fit to the data. Panel b: Nutrient data for all samples used in this manuscript compared to the published P:N relationships of Jones et al. (2008), Dodd et al. (2012), Sutherland et al. (2009). Dashes thick black lines are for Jones et al. (2008), and thin black lines for Sutherland et al. (2009) and Dodd et al. (2012). The dashed red line indicates a best fit to the Bering Strait nutrient data presented here. Note Dodd et al. (2012) uses the same Pacific relationship as Jones et al. (2008) .





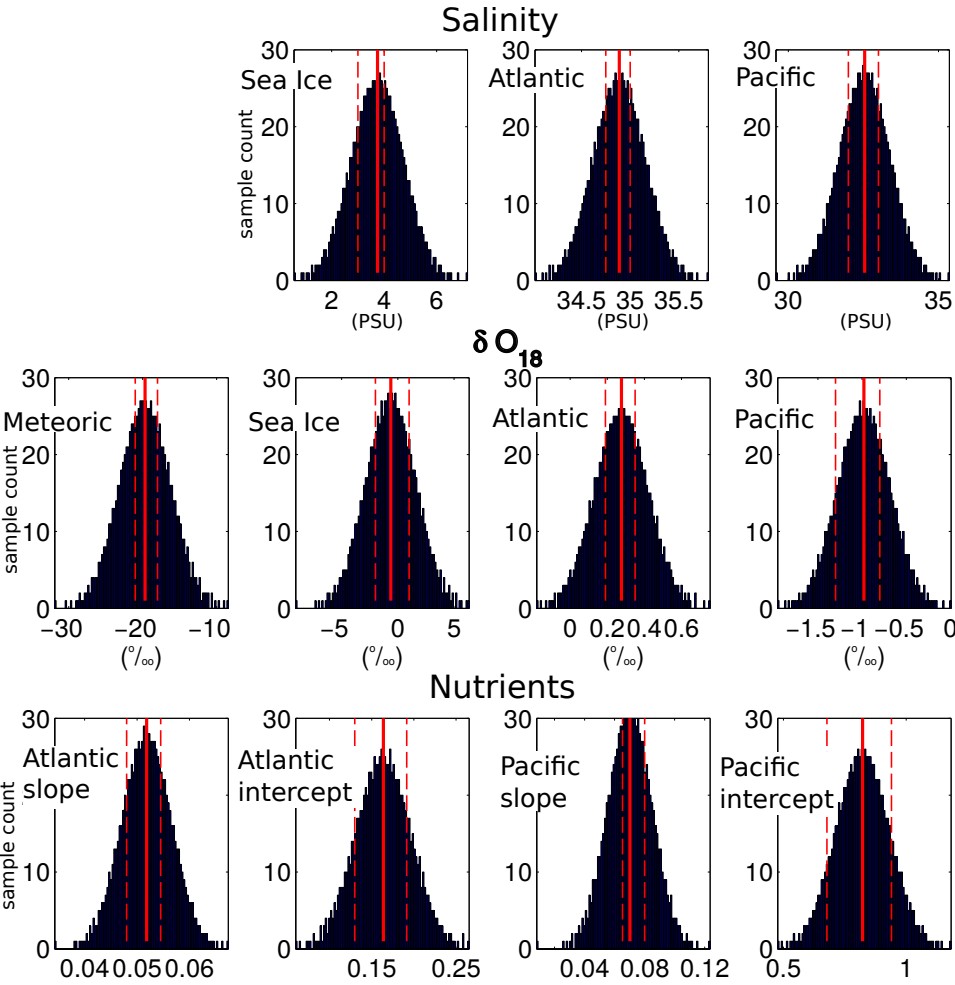

**Figure 4.** Parameter space for the Monte-Carlo simulations. Solid red line indicates the mean of the published values for the parameter; dashed red lines indicate maximum and minimum of published values.





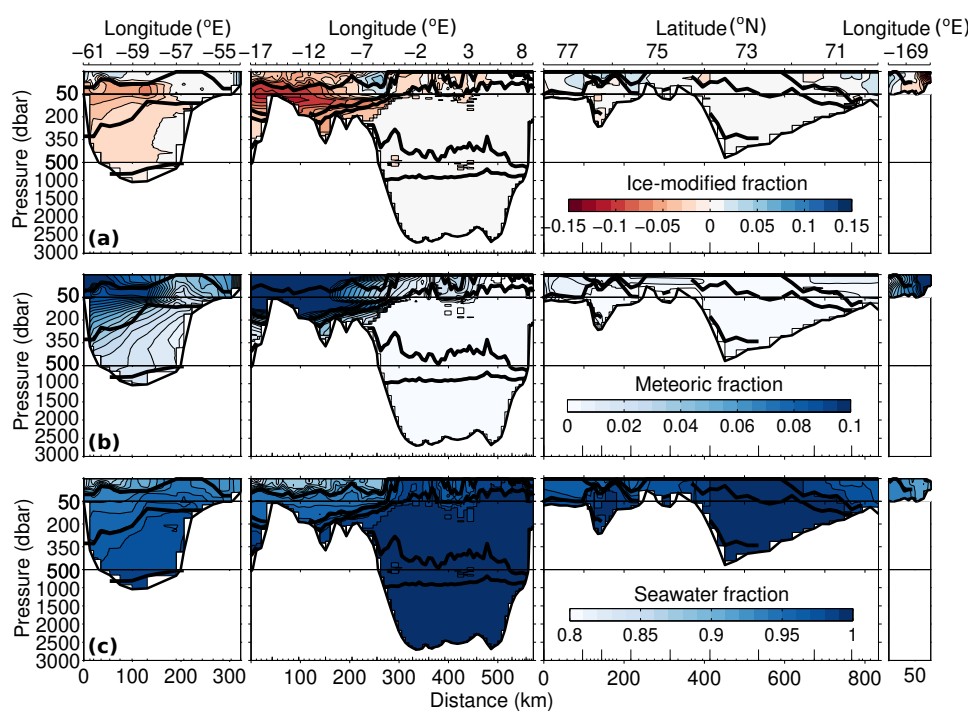

**Figure 5.** Sections of ice-modified fraction (panel a), meteoric fraction (panel b), and seawater fraction (panel c), for the 3EM model, clockwise around the four gateways from Davis to Bering Straits. Solid black lines indicate the isopycnal surfaces separating the main Arctic water masses as described in Tsubouchi et al. (2012). End-members used were the mean of the literature values (see text). Note different color scales for each panel.





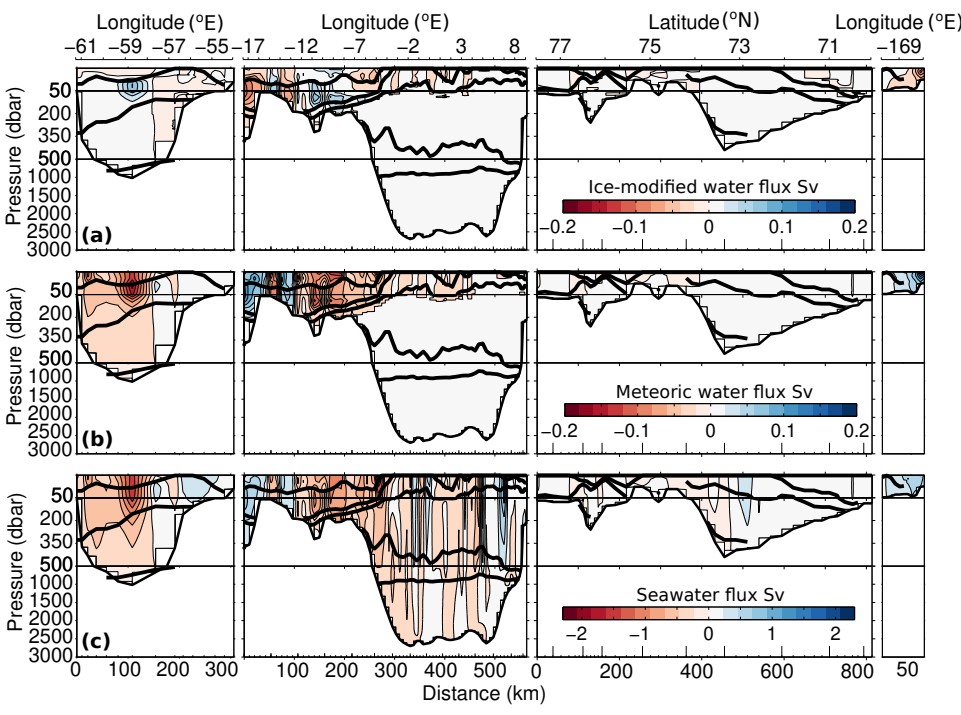

**Figure 6.** Sections of ice-modified water flux (panel a), meteoric water flux (panel b), and seawater flux (panel c), for the 3EM model (Sv), clockwise around the four gateways from Davis to Bering Straits. Solid black lines indicate the isopycnal surfaces separating the main Arctic water masses as described in Tsubouchi et al. (2012). End-members used were the mean of the literature values (see text). Note different color scales for each panel.





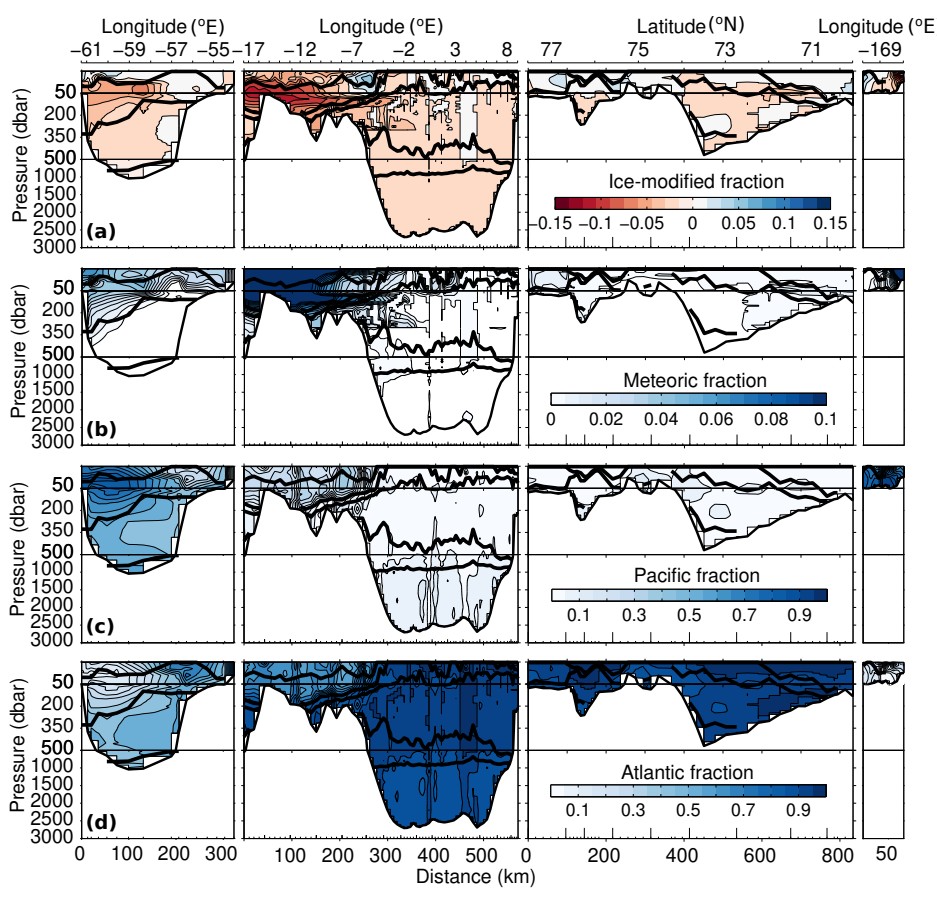

**Figure 7.** Sections of ice-modified fraction (panel a), meteoric fraction (panel b), Pacific fraction (panel c), and Atlantic fraction (panel d), for the 4EM model, clockwise around the four gateways from Davis to Bering Straits. Solid black lines indicate the isopycnal surfaces separating the main Arctic water masses as described in Tsubouchi et al. (2012). End-members used were the mean of the literature values (see text). Note different color scales for each panel.





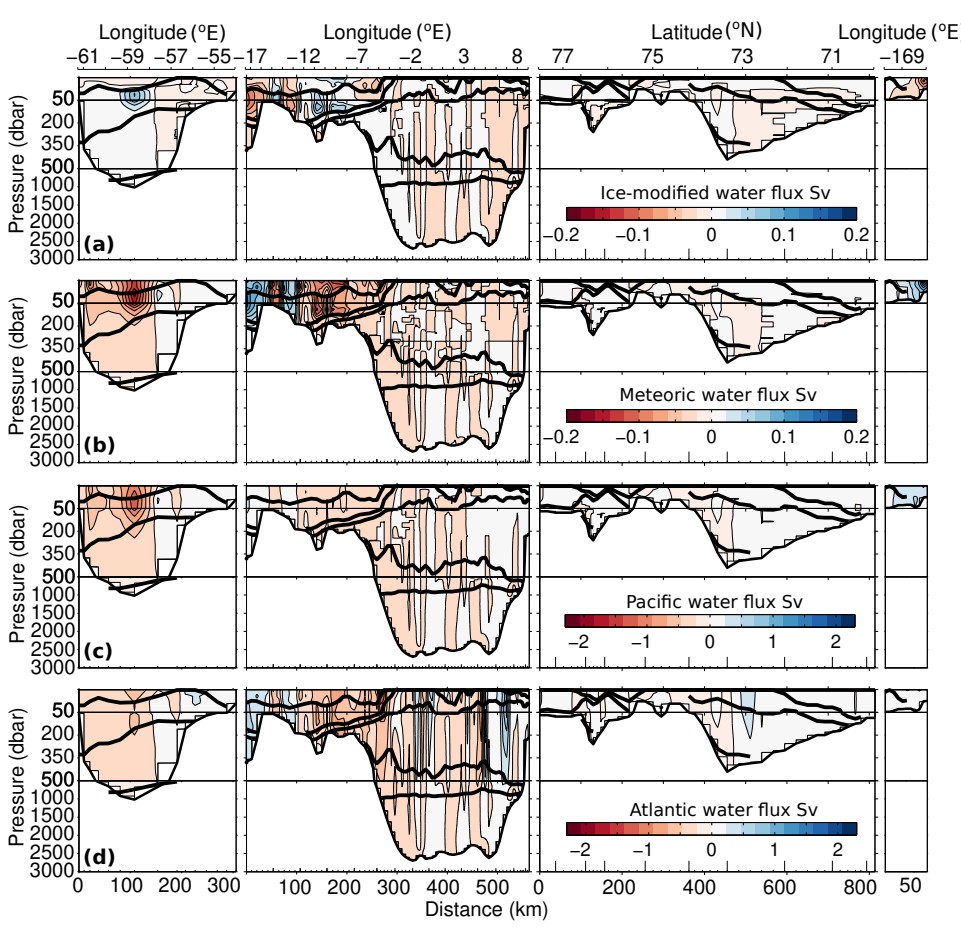

**Figure 8.** Sections of ice-modified water flux (panel a), meteoric water flux (panel b), Pacific water flux (panel c), and Atlantic water flux (panel d), for the 4EM model (Sv), clockwise around the four gateways from Davis to Bering Straits. Solid black lines indicate the isopycnal surfaces separating the main Arctic water masses as described in Tsubouchi et al. (2012). End-members used were the mean of the literature values (see text). Note different color scales for each panel.



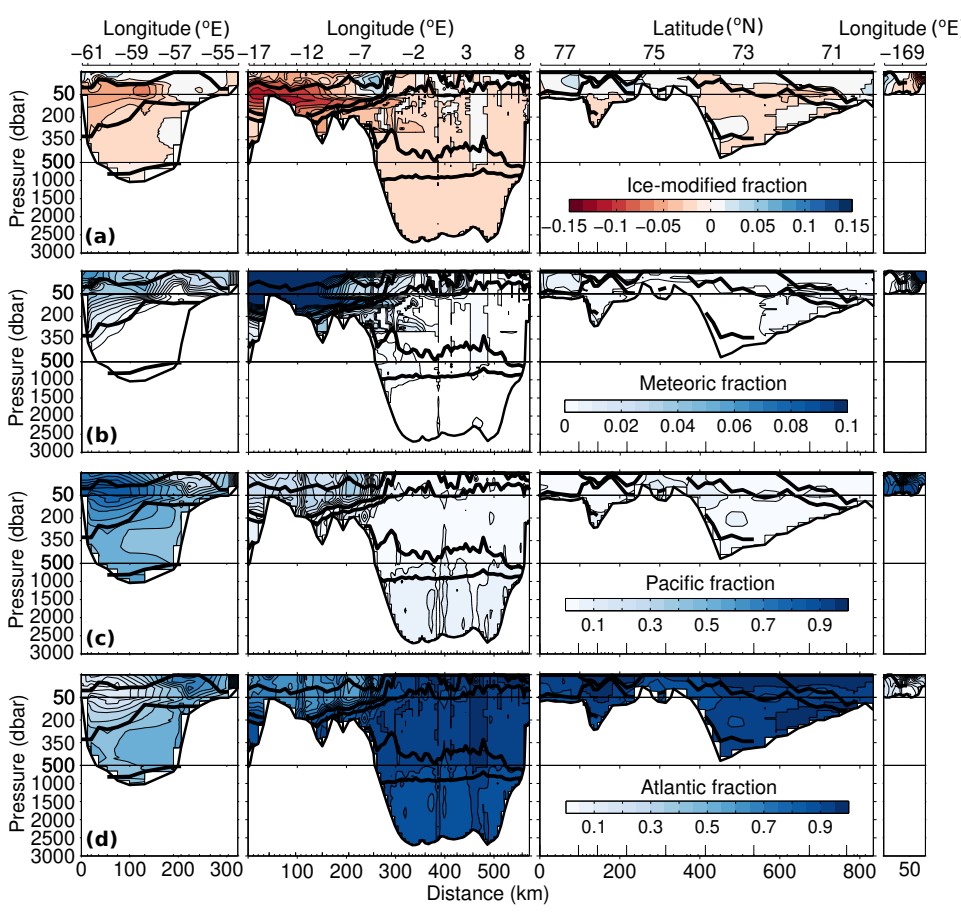

**Figure 9.** Sections of ice-modified fraction (panel a), meteoric fraction (panel b), Pacific fraction (panel c), and Atlantic fraction (panel d), for the 4EM+ model, clockwise around the four gateways from Davis to Bering Straits. Solid black lines indicate the isopycnal surfaces separating the main Arctic water masses as described in Tsubouchi et al. (2012). End-members used were the mean of the literature values (see text). Note different color scales for each panel.



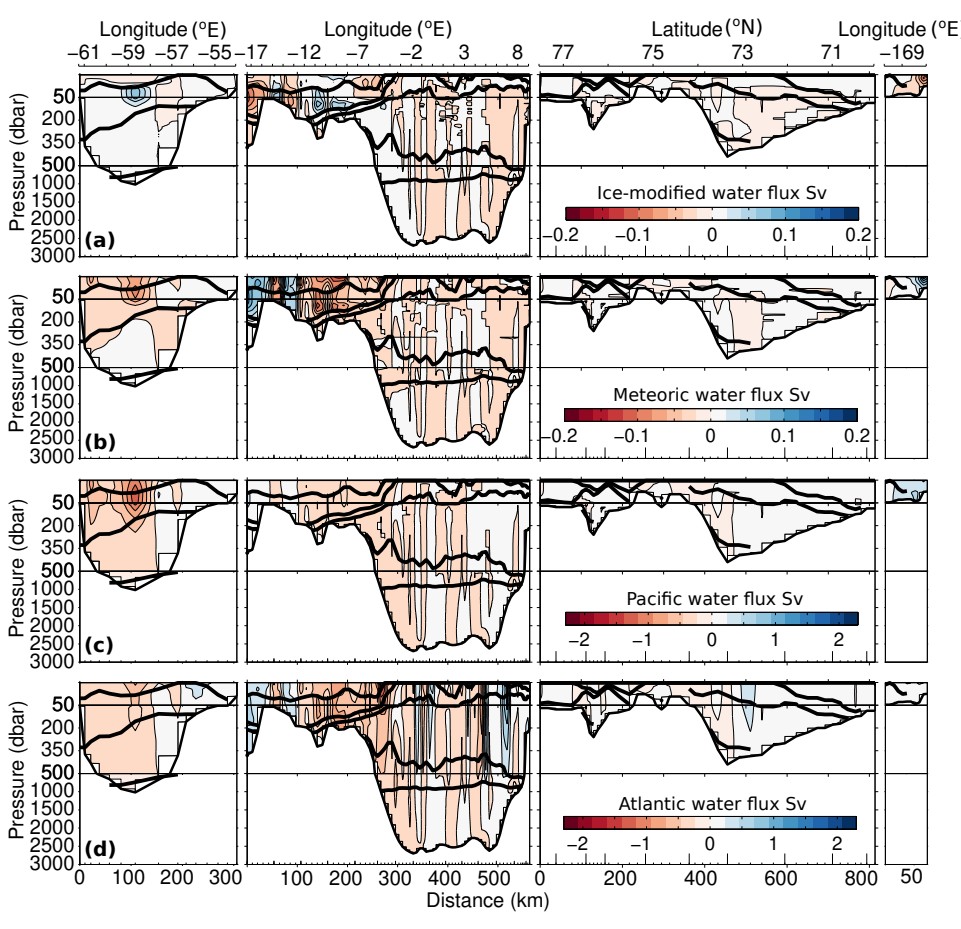

**Figure 10.** Sections of ice-modified water flux (panel a), meteoric water flux (panel b), Pacific water flux (panel c), and Atlantic water flux (panel d), for the 4EM+ model (Sv), clockwise around the four gateways from Davis to Bering Straits. Solid black lines indicate the isopycnal surfaces separating the main Arctic water masses as described in Tsubouchi et al. (2012). End-members used were the mean of the literature values (see text). Note different color scales for each panel.



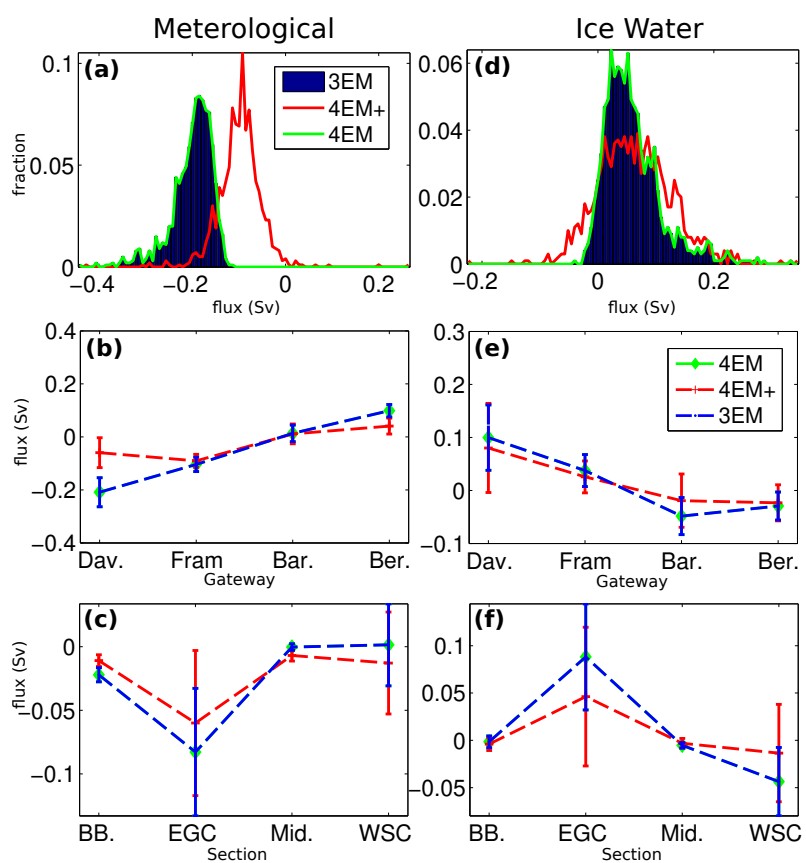

**Figure 11.** Meteoric and ice water volume fluxes. Top row (panels a and d) shows histograms of the total attributed volume fluxes (Sv) for all model schemas. Middle row (panels b and e) shows mean volume fluxes (Sv ± standard deviation) for each gateway. Bottom row (panels c and f) shows volume fluxes (Sv ± standard deviation) for the components of the Fram Strait (Belgica Bank, BB; East Greenland Current, EGC; Mid-strait, Mid.; West Spitsbergen Current, WSC). The 3EM model is in blue, the 4EM model in green, and the 4EM+ model in red. Positive values indicate fluxes into the Arctic.



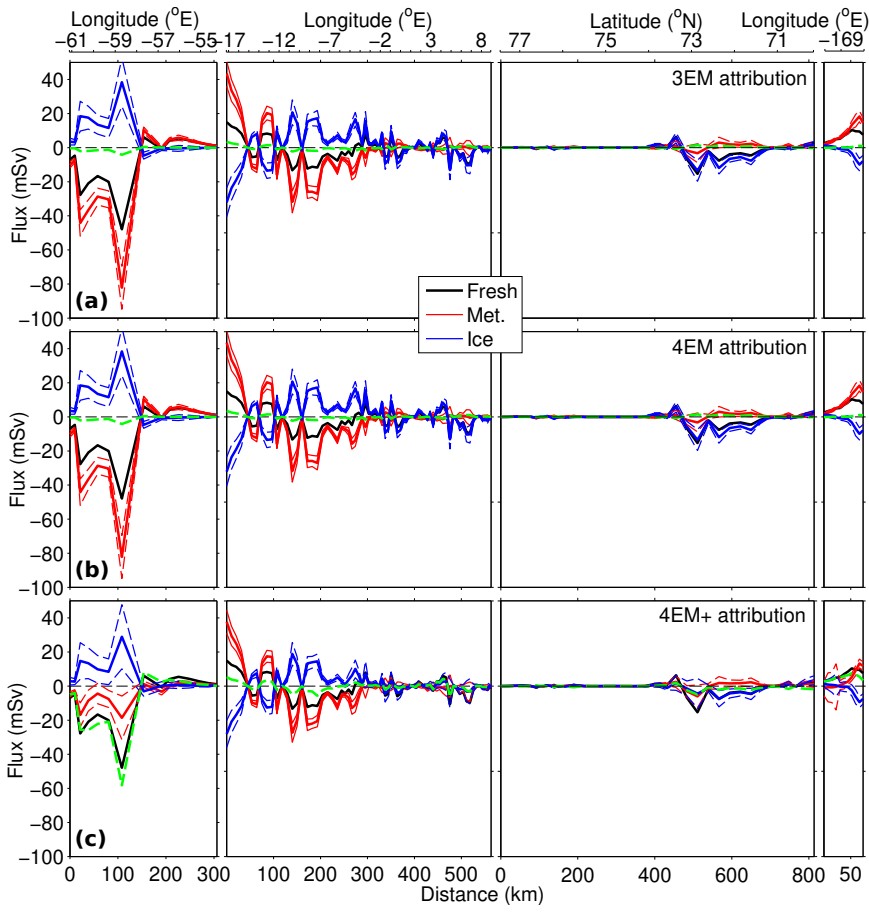

**Figure 12.** Net per-station mean depth-integrated volume fluxes of freshwater from Tsubouchi et al. (2012), meteoric, and ice-modified water components for the three model schemas (panel a 3EM; panel b 4EM panel c 4EM+), clockwise around the four gateways from Davis to Bering Straits. Thick black line shows freshwater flux (mSv), red line meteoric volume flux (mSv ± standard deviation), and blue line ice-modified water volume flux (mSv ± standard deviation). The dashed green line is the difference between the freshwater flux and the sum of the meteoric and ice-modified water fluxes, which is the assumed oceanic freshwater contribution. Positive values indicate fluxes into the Arctic.




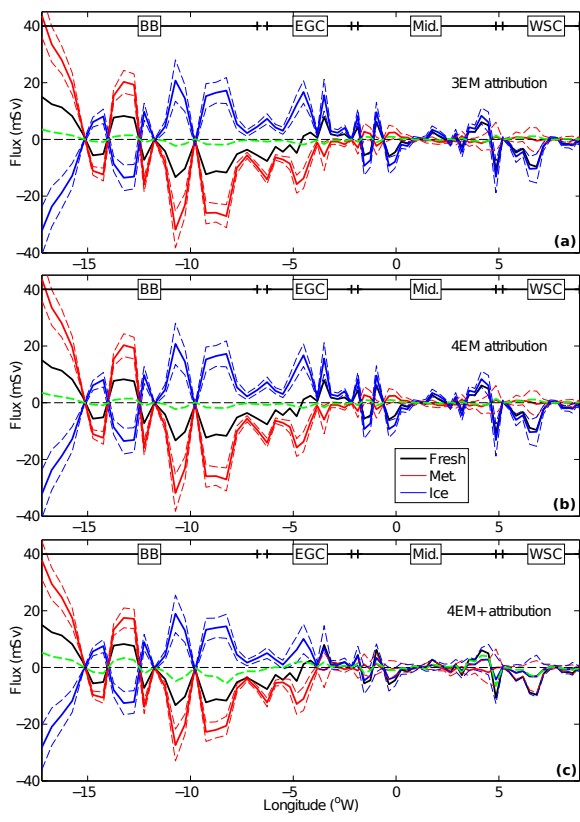

**Figure 13.** Net per-station mean depth-integrated volume fluxes of freshwater from Tsubouchi et al. (2012), meteoric, and ice-modified water components for the three model schemas for the Fram Strait (panel a 3EM; panel b 4EM panel c 4EM+). Thick black line shows freshwater flux (mSv), red line meteoric volume flux (mSv ± standard deviation), and blue line ice-modified water volume flux (mSv ± standard deviation). The dashed green line is the difference between the freshwater flux and the sum of the meteoric and ice-modified water fluxes, which is the assumed oceanic freshwater contribution. Positive values indicate fluxes into the Arctic. Positions of the components of the Fram Strait flux (Belgica Bank, BB; East Greenland Current, EGC; Mid-strait, Mid.; West Spitsbergen Current, WSC) are indicated.



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
