# Peer review of "Manuscript prepared for The Cryosphere Discuss. with version 2015/09/17 7.94 Copernicus papers of the Large X class copernicus.cls. Date: 19 June 2019"

_The Cryosphere, 2018_

## Referee Comment (RC1) · Thomas Armitage (Referee) · 27 Feb 2019

**Review of "Arctic freshwater fluxes: sources, tracer budgets and inconsistencies", by Forryan et al.**

T. Armitage

**General comments**

The authors present a reconciliation the two prevailing methods for determining the freshwater budget of the Arctic Ocean, namely the control-volume based approach (based on salinity and velocity fields at the ocean boundary) and the geochemical tracer approach (where Nitrate/Phosphate ratios and d$^{18}$O are used to determine water origins). I found the paper to be educational and interesting, it is very well-written in general and succinctly conveys its main points. As such my comments are mostly suggestions to improve the presentation. I would note that, while I am broadly familiar with aspects of the Arctic freshwater cycle, I am no expert in geochemical tracers and their interpretation so I cannot fairly judge the merits of that aspect of the paper. From what I have seen I think the manuscript only needs some minor changes before publication. I do question the choice of The Cryosphere as a venue for the paper, since ice only really plays a minor supporting role as a source/sink/vehicle for freshwater. The equivalent EGU journal Ocean Science might be a better choice, or J Physcial Oceanography or JGR: Oceans. But I leave it to the editor to decide in their recommendation.

**Specific comments**

P2L2 and p23L11 – "traditionally divergent" to me implies that the divergence is somehow inevitable, or done on purpose historically. Maybe use "generally divergent" instead.

P2L3 – split the sentence: "…reconcile. The…"

P3L4-10 – I'm not sure the discussion of mid-latitude linkages and AMOC disruption by Arctic FW are really warranted here. Also, my (admittedly limited) understanding of both of these phenomena is that they are highly contentious, and an accurate mention of them would have to also say that some researchers claim there is no evidence that they are occurring or will occur.

P8L5-7 – could you give some indication of the uncertainty associated with the optimal interpolation of the geochemical data?

P9L5 – What is a Redfield nutrient ratio? Certainly my lack of knowledge, but I'm probably fairly representative of the Cryosphere audience…

Figure 2 – (caption) I think the gateways are shown anticlockwise from Davis i.e., Davis, Fram, BSO, Bering? I think you should write on all of these Figures (2 and 5-10) which opening is which, for clarity and ease of interpretation.

P11L4-6 and Figure 3a/b – the fits to the green points (Fram) are poor, or rather the data are clearly not linear, which you attribute to the presence of Greenland Ice Sheet meltwater. Is there a way to exclude the Greenland water masses from your data in order to improve the fits? The linear regression to the green points in fig 3A especially is clearly not suitable and shouldn't be used for further analysis.

P13L25 and Figure 6 – is flux positive in or out/positive or negative? Say explicitly early on for ease of interpretation.

P14L1 – "positive fluxes indicating an export of high-salinity waters", is this equivalent to an import of freshwater? Seems more intuitive to talk about freshwater fluxes as that's the main focus of the paper.

P14L11-17 and Tables 3-8 – I find reading data off tables pretty unhelpful in general, but especially when we are trying to compare data between different tables as here. I think you could easily summarise the budgets presented in tables 3-8 in one figure with multiple panels, or in a couple of separate figures. Personally I would use bar charts with error bars, and you could also include the Fram Strait break down as 'sub-bars' of the Fram bar. Would highly recommend this as it would make interpretation/comparison between the model runs much easier.

P14L27 – the ice-modified water in the WSC is from recirculation, right? State this here

P15L26 – "large uncertainty", the uncertainty is actually smaller than for the 3EM/4EM models, the relative uncertainty is larger though if that's what you mean.

Section 4.1 – Is the apparent consistency between the 3EM and 4EM models a surprise given that the difference between them is just the use of the geochemical data to partition the seawater into Atlantic/Pacific fractions? In Figures 11, 12, and 13 I can see no difference between the 3EM/4EM fluxes. In other words, are the 3EM/4EM fluxes consistent just by construction? If so you should say this, as it is misleading to say they are "consistent" when they are simply the same by construction. Perhaps my misunderstanding.

P22L8-12 – while you cannot ascertain exactly the source of this water transformation, could you speculate at all? At least on the classes of processes that might cause this?

Section 4.3 – I was wondering if you could also provide a paragraph with some perspectives on 1) future research using these methods/datasets, and 2) implications for Arctic Ocean climate monitoring in terms of observation systems and optimal approaches at analysis/modelling.

---

## Short Comment (SC1) · 5 Apr 2019

Dear authors,

I have a couple of comments on your paper.

My first comment regards the implementation of the N:P ratio method used to identify Pacific Water. Figure 7 in your paper shows low fractions (10 – 20 %) of Pacific Water along much of the boundary section in places that seem unlikely. For example, much of Fram Strait is filled with low fractions of Pacific Water below 1000 m. Pacific Water is buoyant and enters the stratified Arctic through a 60 m deep channel, so it seems unlikely that Pacific Water should be found at the bottom of Fram Strait. I think these apparently-spurious Pacific Water fractions might need be addressed before we can

expect good results from the inverse model. There are a couple of ways in which this might be achieved:

1) Some of the apparently-spurious low Pacific Water fractions might arise from uncertainties in the end-member properties. If the low fractions are not significantly different from zero it might be justifiable to suppress them.

2) An alternative approach could be to apply the N:P technique only in the depth range where Pacific Water is likely to be found, assuming fractions below some depth threshold to be zero. The N:P ratio method has a large errors associated with it and if it is applied indiscriminately over large areas where we would not expect to find Pacific Water the accumulated systematic errors probably become quite significant.

However, the limitations of the N:P ratio technique are perhaps not the main reason that the inverse model does not balance for Pacific Water. My second comment regards the application of the inverse technique to Pacific Water in the Arctic Ocean. I'm not very familiar with inverse modelling, but I think the technique assumes that the system is in a steady state. The repeated Pacific Water sections in Dodd et al., 2012 (cited in your paper) indicate that the flow of Pacific Water through the Arctic is not in a steady state. At least in Fram Strait, Pacific Water is released in pulses with peak Pacific Water fractions of up to 80 % interspaced with periods where peak Pacific Water fractions barely exceed 20 %. The duration of pulses is probably of the order of 2 years, which is quite short relative to the time required for Pacific water to cross the Arctic. I'm not exactly sure how this can be best addressed, but I think the paper should at least discuss this steady-state issue.

One reason that the inverse model might balance for salinity/freshwater, but not for Pacific Water, could be that in years when Pacific Water is not present in a given location it tends to be replaced by another halocline water mass of similar density (ie: rather similar salinity).

There is some denitrification in the Arctic and I agree that when using the N:P ratio

technique, some Atlantic Water will apparently be transformed into Pacific Water over the shallow shelves. That is indeed a fundamental limitation of the technique. However, if the steady-state issue is as serious as I think it is, then I'm not sure that the results of the inverse model give us much new information about the reliability of the N:P technique. Please do correct me if I am wrong about something here though!

Incidentally, I'm inclined to agree with the first reviewer's suggestion that this paper might be better suited to Ocean Science Discussions than The Cryosphere.

Regards,

Paul A. Dodd

---

## Referee Comment (RC2) · Wilken-Jon von Appen (Referee) · 9 Apr 2019

Review of A. Forryan et al "Arctic freshwater fluxes: sources, tracer budgets and inconsistencies" by Wilken-Jon von Appen

The paper combines several different lines of investigation that have existed in the literature and that have come to partially contradicting results. The authors demonstrate where these contradictions stem from. As a result, this paper is a major advance in the understanding of the cycle of H2O and of salt in the Arctic Ocean. This is based on data from summer 2005 and it is implied that that data represents a steady state situation. This is the right first step, but it should be pointed out as such with a discussion of the implications of the steady state assumption. Nonetheless, the resulting numbers

are useful to the oceanographic community. Furthermore, it is demonstrated that P\* (calculated from measured nitrate and phosphate concentrations) is not really useful to distinguish "Atlantic" from "Pacific" waters in the Arctic. This points to the order of magnitude of non-Redfield ratio biogeochemical processes in the Arctic Ocean whose order of magnitude cannot otherwise be observed directly. There are, however, also a few issues with this paper, in particular with respect to "freshwater". These mostly concern Figures 12/13. As I describe in detail below, I would leave those out. The manuscript is well written and straightforward in its presentation. But I have a few suggestions, which I think could improve the clarity of the manuscript ("what exactly are the authors doing at each step in the manuscript and why?"). Therefore, I would definitely like to see this manuscript published, but only after a major revision. Furthermore, while sea ice is important in this paper, it is clearly within the discipline of oceanography (physical and chemical). Therefore, I would suggest publication in Ocean Science rather than The Cryosphere. I encourage the editors and chief editors of the two journals to see whether there is a way to transfer the manuscript without restarting the peer review process from scratch.

Line by line comments

p2l7 "(liquid) freshwater fluxes" and p2l23 "freshwater" Please give a clear definition of what you mean by freshwater. Is this H2O? At this point there are too many different (sometimes meaningful) definitions in the literature that you cannot assume the readers know exactly which one you are using here.

p4l12-14 This equation appears to only hold for 1 constant salinity at the inflow and another constant salinity at the outflow from the box. For the Arctic Ocean, that is clearly not given. To me it is not clear from this manuscript or from Bacon et al 2015 whether Sbar is an area mean or a transport weighted mean salinity over the boundary. I would appreciate it if the authors could clarify this here.

p5l2-3 and l6-7 "accurate estimates of freshwater flux require the definition of an appro-
priate reference salinity (Sbar)" and "the boundary-mean salinity is the only appropriate reference salinity" I do not think that either of these sentences is correct. But rather than arguing over whether they are correct, I would suggest to leave them out as they are in fact not crucial to anything that follows later in the manuscript.

p5l29 Nd isotopes and REEs have also been used as conservative tracers of different rivers in the Arctic Ocean, e.g. doi:10.1016/j.gca.2016.12.028

p6l14-17 Again, while this is an appropriate step to take at this point (and better results might not be obtained from data at this point), it should still be pointed out that this is not perfect and there are in fact possibly large systematic errors arising from the sampling locations and/or spatially (potentially) insufficient sampling. It would be nice to mention these points with at least a few sentences.

p7l8 The correct statement would be that this "conserves volume and salt transports", not that it "conserves volume and salinity transports"!

p7l14 Please say what you mean by the plus/minus here. E.g. it could be standard deviation or standard error.

```
p7l15 "1.0 +- 0.2" not "1 +- 0.2"
```

p7l18 "Sv" not "sV"

p7l24 Please state where your information on sea ice export is from, e.g. satellite observations of sea ice drift and sea ice volume?

p8l5 "nutrient and delta18O data were optimally interpolated" Comment on whether the spatial distribution of the data was sufficient or whether there could be interpolation issues.

p8l9 "grid cells as hydrographic stations" I see where this is coming from, but it is still a strange way to formulate it.

p8l15-27 For me this was totally incomprehensible upon first reading. The terms "3EM",

TCD
"4EM" and "4EM+" are not self explanatory. I would strongly advise to make a diagram or a table. A suggestion would be a table like this (columns could not be formatted in plain text, so the individual lines of the table are grouped together):

Model name

Constraints

End members that are solved for

Comments

new line

3EM

Volume conservation, salinity data, delta180 data

Seawater fraction, meteoric water fraction, ice melt water fraction

Seawater is water with S=35 irrespective of whether it enters from the Atlantic or the Pacific

new line

4EM

Volume conservation, salinity data, delta180 data, P\* data

Atlantic water fraction, Pacific water fraction, meteoric water fraction, ice melt water fraction

Atlantic water and Pacific water are defined to have identical S and delta18O end member characteristics, but different  $P^*$

new line

4EM+
Volume conservation, salinity data, delta180 data, P\* data

Atlantic water fraction, Pacific water fraction, meteoric water fraction, ice melt water fraction

Atlantic water and Pacific water are defined to have different P\* and similar, but not identical S and delta18O end member characteristics

I would also already add a sentence like the following one here, because (anyways for me) it was not clear why you do these two versions with 4EM and 4EM+: "4EM+ is degenerate (meaning that numerical values are strongly affected by small perturbations) because the distinct source salinity and delta18O values of Pacific water are on a mixing line between the meteoric and Atlantic Water end member quantities.

p9l14 1 sentence here why you use Pest: in order to judge the method, not for use in the method itself

p10I5 Again, I think this would be much clearer if you could refer to the table as I suggested above.

p11l14 refer to Table 2 in this sentence

p12l16 Again, area mean or transport weighted mean?

p14l1 m/s (see my comment below on Figure 6)

p14l11 It might be helpful to remind the reader that your +- values stem from the Monte Carlo simulations.

p15l2 Also, here 1 sentence would be in order repeating what the difference between 4EM and 4EM+ is and why you do both calculations.

p15l22-23 and l25-26 Please don't just show both sets of numbers, but also comment on which one you think makes more sense.

p16l16 Add a sentence such as: "Both of these numbers should be approximately 0 and
therefore, we consider this a model/methodological/data(?) mistake for the following reasons..."

p16l21 You are only looking at data from 1 summer month. Discuss whether all of this should be balanced in the quasi-synoptic view of the data you use.

p17l16-18 Neither of these views seems plausible for the West Spitsbergen Current. Should the Atlantic water salinity not rather match the WSC closely?

p18l14-15 Should this not be considered everywhere?

p19l2 "solid (sea ice) fraction" instead of "solid, sea ice, fraction"

p19l27 "(at least when considering full depth assessments)" It is not clear why that caveat is necessary and why the sentence is not correct without the added information in brackets.

p2016 How can a river be a sink? Processes on the continental shelf near the river could be sink processes.

p20l20 no ":"

p20l25 Explain how I would see that from Figure 3 and what degenerate means in that context.

p21l1 "boundary mean salinity" Again, where do I "see" that in Figure 3?

p21l6 Refer to Table 7

p21112-24 This text and the associated Figures 11-13 should in my opinion be removed from the manuscript as it is unclear what you mean by "oceanic origin freshwater". Additionally, there is no insightful information contained in them.

p22I17 "Carmack et al. (2016, Appendix)"

p23l6 "salt conservation"!
Tab1 Why is there a larger line break after the first line of delta18O and salinity?

Tab1 2nd line under ice melt: What is "surf"?

Tab3 Similar to p7l24, where is the information about -0.040Sv solid ice melt from?

Fig2 I think the other piece of interpolated data that your study is based on is crosssectional velocity. I would recommend to add this as a top (4th) panel to Fig2. In that case the reader does not need to refer back to TB12 to get that information

Fig2 caption I1 "P\*" should be with a superscripted "\*"

Fig2 caption I4 Repeat what the main Arctic water masses are so that the reader does not need to refer back to TB12.

Fig2 caption Add: "Note the broken scaling of the y-axis."

Fig3 Your 3EM model solves the classical end member decomposition in the triangle that is drawn in panel a. Your 4EM models essentially are the same, only that they solve the end member decomposition in the tetrahedron that would result if you were to extend panel a in the vertical with the vertical axis being P\*. Since you can't add a 3 dimensional figure to the paper, I would recommend to at least add plane views of this tetrahedron with the data and dashed lines plotted into the panels just as you are doing in panel a right now. Common axes can be aligned with each other. My suggestion: 4 panel figure. top left panel as your panel a. top right panel x-axis P\* y-axis delta18O, bottom left panel x-axis salinity y-axis P\*, bottom right panel your current panel b. Also please substitute the current legend in panel b by a legend for the dashed lines and comment in the figure caption that all symbols and lines are the same in all panels. The 18 in the ylabel of panel a should be superscripted not subscripted.

Fig3 caption I4 "Dashed thick"

Your units in Figs 6/8/10 and 11a/b are wrong. They should be "Sv/m/km" or more conventionally "m/s". Note that you only arrive at units of transport (Sv) after integrating

TCD
the data in the figures in the horizontal and vertical dimensions. Same applies for Figs 12/13 where your units should be m2/s or Sv/km or similar.

Fig7 What is plotted in panels 7a and 7b is different from what is plotted in panels 5a and 5b, yet the values in the Met. and Ice Melt columns of Tables 3 and 5 are identical. In my opinion, only either one of those can be correct.

Fig11/12/13 can in my opinion be deleted from the manuscript. One of the reasons is that I do not understand what the black line in Fig12/13 is supposed to be.

Fig12/13 (if you keep them) Add green line to legend.

p48l14-16 A correct reference to this data publisher contains the complete DOI and it is not a tech report. Compare how the citation is provided on the webpage of the data set. In addition, you have the wrong title which means that it took me some time to find the data set you are referring to! "Kattner, Gerhard (2011): Inorganic nutrients measured on water bottle samples during POLARSTERN cruise ARK-XXI/1. PANGAEA, https://doi.org/10.1594/PANGAEA.761684"

TCD

---

## Author Response (AR1)

**Replies to reviews of Forryan et al.** "Arctic freshwater fluxes: sources, tracer budgets and inconsistencies".**

Tom Armitage (TA) and Wilken-Jon von Appen (WA) provided reviews of our manuscript, and also Paul Dodd (PD) wrote an interactive ("short") comment. These three will be well aware that not all reviews are helpful. In the present case, all three caused us to examine certain matters more closely, with the result that our Section 4 (Discussion and Summary) has been overhauled and is now quite different, and there are major edits elsewhere. We also became conscious that aspects of our use of language were opaque in some cases, in particular around: ice-modified waters, where we now simplify to "sea ice melt water" and "brine"; the distinction between what we now call "oceanic water", meaning the complex of all components, distinct from ocean (Atlantic / Pacific) seawater sources, which we now call "seawater"; and the terminology around "freshwater fluxes" and also around methods is now made explicit. The relevant material appears in Sections 1 and 2, and specific instances are detailed below. We believe that the manuscript is significantly improved, and we express our sincere thanks. We reply to specific comments (in italic font) below, after first replying to a general point.

**Regarding choice of journal**

We selected The Cryosphere for a number of reasons. On The Cryosphere's home page (https://www.thecryosphere.net), it is stated that the journal is "dedicated to ... all aspects of frozen water". Furthermore, The Cryosphere remit includes publishing articles "including studies of the interaction of the cryosphere with the rest of the climate system". The Cryosphere is about more than ice – it is about the cryosphere, which is what has made it a distinctive and excellent vehicle for the publication of articles about the cryosphere since its foundation over a decade ago. As a relatively early example of a paper that took an approach analogous to ours, we cite Serreze et al. (2009, https://doi.org/10.5194/tc-3-11-2009), entitled "The Emergence of Surface-Based Arctic Amplification.", and which is "about" the Arctic atmosphere, while treating sea ice as an essential component of the Arctic climate system. This article has been cited over 400 times since its publication. As another instance, we note the publication by TA and one of us (SB) of an article in The Cryosphere in 2017 entitled "Arctic Ocean surface geostrophic circulation 2003–2014" (https://doi.org/10.5194/tc-11-1767-2017), which is mainly "about" the (liquid) Arctic Ocean.

In the present case of our manuscript, sea ice is a fundamental component of our analysis. It appears *per se* as a key element of the net freshwater budget of the Arctic ice and ocean system, and without the impression made by sea ice processes on the oceanic concentrations of oxygen isotopes, our analysis would not function. We believe that by presenting our work through The Cryosphere, our article will directly reach the audience that can best appreciate it.

**Reviewer 1 (TA)**

P2L2 and p23L11 – "traditionally divergent" to me implies that the divergence is somehow inevitable, or done on purpose historically. Maybe use "generally divergent" instead.

Done

P2L3 – split the sentence: "...reconcile. The..."

Done

P3L4-10 – I'm not sure the discussion of mid-latitude linkages and AMOC disruption by Arctic FW are really warranted here. Also, my (admittedly limited) understanding of both of these phenomena is that they

are highly contentious, and an accurate mention of them would have to also say that some researchers claim there is no evidence that they are occurring or will occur.

These references are included as part of the introduction to inform the reader of the potential wider impact of changes in the Arctic. Both the relevant statements have caveats (underlined below):

P3 L12 Changes in the Arctic heat budget may affect the strength ...

and

P3 L16 Arctic freshwater export also has the potential to change ...

These statements are uncontentious and we prefer to leave them unchanged.

*P8L5-7 – could you give some indication of the uncertainty associated with the optimal interpolation of the geochemical data?*

The section covering optimal interpolation has now been revised.

**P8 L22** The  $\delta^{18}$ O and nutrient data were optimally interpolated (Roemmich, 1983) vertically in pressure and horizontally in distance to match the TB12 model domain (Fig. 2). The interpolation recovers the measurements for each sample point and interpolates between values to fill the unsampled areas of the domain. The resulting nutrient fields show typical features, including low concentrations in the upper, sunlit layers as a consequence of nutrient utilisation during primary production, and concentrations that increase with depth due to remineralisation and/or dissolution of sinking particles; see also Torres-Valdés et al. (2013). The  $\delta$  18 O sample resolution is mainly adequate to capture the significant Arctic Ocean features, although in the Fram Strait section around 6 ° W, there is only a single station to represent the East Greenland Current, so that horizontal gradients to either side of this station will only be approximate, therefore.

P9L5 – What is a Redfield nutrient ratio? Certainly my lack of knowledge, but I'm probably fairly representative of the Cryosphere audience...

A fair point. A gloss on the Redfield ratio is now included:

**P5 L27** Concentrations of dissolved inorganic nutrients in seawater and the elemental composition of phytoplankton populations are observed to occur at broadly the same stoichiometric rations (Redfield et al., 1963). Where nutrient availability does not limit phytoplankton growth, this indicates that the ratio of the uptake of nutrients (the ratio of nitrate to phosphate, in this case) by phytoplankton, known as the "Redfield ratio", is fixed. In the Arctic context, this implies that deviations from typical Redfield ratios of seawater concentrations of these inorganic nutrients may serve as tracers of the geographic origin of seawaters, which would be useful to understand seawater pathways through the Arctic Ocean.

Figure 2 – (caption) I think the gateways are shown anticlockwise from Davis i.e., Davis, Fram, BSO, Bering? I think you should write on all of these Figures (2 and 5-10) which opening is which, for clarity and ease of interpretation.

All the relevant figures have been updated to include labels on the gateways.

P11L4-6 and Figure 3a/b – the fits to the green points (Fram) are poor, or rather the data are clearly not linear, which you attribute to the presence of Greenland Ice Sheet meltwater. Is there a way to exclude the Greenland water masses from your data in order to improve the fits? The linear regression to the green points in fig 3A especially is clearly not suitable and shouldn't be used for further analysis.

The linear fit lines originally plotted on the inset to Figure 3a were intended to give an indication of how closely these data conform to the mixing lines of the plotted end-points, and to demonstrate that the data used are representative, meaning that the calculated end point fell within both the range of the literature end-point values and within the end-point distribution used for the Monte-Carlo simulations. The deviations exhibited by the Fram section are explained in the text (as noted). The linear regression values are not used, outside this, in any analysis. To avoid confusion, the regression lines have now been removed from the figure. Figure 3 caption now reads:

**P41** Panel a: Salinity –  $\delta^{18}$ O relationship for all samples used in this manuscript; mean literature end-points (± standard deviation) are marked. Red crosses indicate the mean values of literature end-points and black dashed lines the mixing lines between them. Panel b: Nutrient data for all samples used in this manuscript compared to the published N:P relationships of Jones et al. (2008), Dodd et al. (2012), Sutherland et al. (2009). The dashed red line indicates a best fit to the Bering Strait nutrient data presented here. Symbols denoting the data from each section are common to both panels. Note Dodd et al. (2012) uses the same Pacific relationship as Jones et al. (2008).

P13L25 and Figure 6 – is flux positive in or out/positive or negative? Say explicitly early on for ease of interpretation.

The statements have now been clarified as suggested. Figures 5 and 6, **Tables 4–13**, Positive values indicate flux into the Arctic.

P14L1 – "positive fluxes indicating an export of high-salinity waters", is this equivalent to an import of freshwater? Seems more intuitive to talk about freshwater fluxes as that's the main focus of the paper.

While what you say is true, we frame this point in terms of the physical process (export of high-salinity waters) and not in terms of the impact on the freshwater flux calculation, because we are describing fluxes of source fractions. For this reason, we prefer to leave the text unchanged.

P14L11-17 and Tables 3-8 – I find reading data off tables pretty unhelpful in general, but especially when we are trying to compare data between different tables as here. I think you could easily summarise the budgets presented in tables 3-8 in one figure with multiple panels, or in a couple of separate figures. Personally I would use bar charts with error bars, and you could also include the Fram Strait break down as 'sub-bars' of the Fram bar. Would highly recommend this as it would make interpretation/comparison between the model runs much easier

We have a lot of sympathy with the sentiment because there is no doubt that tabulation is cumbersome. However, bar charts are much worse because the range of volume flux values is so large – three orders of magnitude – so that small but important freshwater fluxes (of some tens of mSv) become invisible. Therefore we leave the tables as they are, but this comment did prompt us to expand a little on the explanation of the approach and method in Section 2.3, to illustrate the origins of this range in various physical processes apparent in (for example) Figure 3, panels (a) and (b).

P14L27 – the ice-modified water in the WSC is from recirculation, right? State this here

There are several  $\delta^{18}$ O samples in this area that lead to the apparent presence of ice modified water (brine). Closer examination of our results and also of source water properties (Frew reference below) led us firstly to make a new run of the 3EM model, and secondly to revise our reasoning and text. In the new Section 4.1, we write as follows.

**P20 L 1 Section 4.1** The models generate apparent brine imports in the WSC and the Barents Sea Opening, both of magnitude ~ 45 mSv, a total of ~ 90 mSv with a large relative uncertainty of ~ 50 mSv. If correct, this is a substantial component of the Arctic Ocean freshwater budget. These (apparent) fluxes are too small to be visible on Fig. 5, but for scale, note that each new (oceanic water) inflow is ~ 3 Sv, 1 % of which is 30 mSv. These brine fluxes are consequences of weakly positive  $\delta^{18}$ O anomalies centred around ~ 300 m depth in both locations, each about 200 m thick and each spanning ~ 200 km. The presence of these features in both Fram Strait and the Barents Sea Opening suggests that they are source water (Atlantic seawater) properties and not the result of modifications by local processes. Frew et al. (2000) examine the oxygen isotope composition of northern North Atlantic water masses from measurements made in 1991. Considering the waters of interest here – the upper ~ 500 m in the eastern North Atlantic (their stations 10, 24, 26, 72) – we find (broadly) salinities and  $\delta^{18}$ O values in the ranges 35.0 - 35.2 and 0.2 - 0.4 ‰orespectively (their Fig. 2). This combination and range describes the part of the dense cloud of points heading a short distance "north-eastwards" in phase space away from the seawater endpoint (Fig. 3 panel a inset).

A consistent interpretation of the apparent WSC and Barents Sea Opening brine imports, therefore, is that they are actually manifestations not of local processes but rather of source water variability, in the light of our salinity (34.662) and  $\delta^{18}$ O (mean 0.2 ‰) endpoints.

P15L26 – "large uncertainty", the uncertainty is actually smaller than for the 3EM/4EM models, the relative uncertainty is larger though if that's what you mean.

The statement has been revised - what we meant was:

P17 L12 smaller with a large relative uncertainty

Section 4.1 – Is the apparent consistency between the 3EM and 4EM models a surprise given that the difference between them is just the use of the geochemical data to partition the seawater into Atlantic/Pacific fractions? In Figures 11, 12, and 13 I can see no difference between the 3EM/4EM fluxes. In other words, are the 3EM/4EM fluxes consistent just by construction? If so you should say this, as it is misleading to say they are "consistent" when they are simply the same by construction. Perhaps my misunderstanding.

The difference between 3EM and 4EM models is the use of inorganic nutrients to attempt to discriminate between seawater of Atlantic and Pacific origins, which we now state as such (P16 L19).

P22L8-12 – while you cannot ascertain exactly the source of this water transformation, could you speculate at all? At least on the classes of processes that might cause this?

Torres-Valdes et al. (2016) tested one hypothesis regarding the possible role of dissolved organic nutrients, only to eliminate that option. Their final section comprises a logical conspectus of research avenues to pursue to resolve the problem. We do not wish to repeat that text, but we have written a new Section 4.2 on "Pacific" water. The new text begins by devoping a hypothesis around denitrification (a new paper Matthew Alkire was very useful) as below, and continues, using the suggestion by WA (see below) to look at recently-published material on more exotic tracer species, which supports our development of the hypothesis into dense water formation in the Barents Sea. Section 4.2 begins as below; refer to the manuscript for the full text.

**P 23 L1** A credible hypothesis to explain all these observations – the doubling of Pacific export over import, the transformation of Atlantic water, and the deep presence of Pacific water – concerns denitrification, the process that occurs in ocean sediments and removes nitrate from the ecosystem by discharging N2. Chang and Devol (2009) estimate a net pan-Arctic denitrification rate of ~ 13 Tg-N yr-1, with much of that expected to occur in the shallow waters of the Barents and Chukchi Seas (6 and 3 Tg-N yr-1 respectively). They further note the likelihood that the process is a consequence of sea ice retreat enabling increased primary production through increased shelf-break upwelling, which delivers nutrient-rich waters to upper-ocean waters with greater light availability; the resulting increase in export production then fuels higher rates of sedimentary denitrification.

Section 4.3 – I was wondering if you could also provide a paragraph with some perspectives on 1) future research using these methods/datasets, and 2) implications for Arctic Ocean climate monitoring in terms of observation systems and optimal approaches at analysis/modelling.

We have re-written Section 4.4 Perspectives, which now concludes as follows.

**P26 L23** We envisage that sustained measurement of suitable tracers around the Arctic boundary has the potential to further our quantification and understanding of key processes, variability and timescales and to help mitigate the scarcity of observations in the Arctic Ocean interior. More (and more reliable) tracers are needed, more observations of more "traditional" tracers are needed through the water column (from surface to sea bed), more of those observations are needed in seasons outside summer-autumn, and we need better understanding of Arctic Ocean biogeochemical processes.

**Reviewer 2 (WA)**

p217 "(liquid) freshwater fluxes" and p2123 "freshwater" Please give a clear definition of what you mean by freshwater. Is this H2O? At this point there are too many different (sometimes meaningful) definitions in the literature that you cannot assume the readers know exactly which one you are using here.

We agree with the reviewer's sentiment here. Freshwater flux is now defined in the introduction.

**P3 L 20** We define a flux of freshwater to mean the rate of addition of pure water to (or its removal from) the ocean surface, by exchanges with the atmosphere (evaporation [ E ] and precipitation [ P ]) and by input from the land (runoff [ R ]). The total ocean surface freshwater flux F is then F = P - E + R.

p4l12-14 This equation appears to only hold for 1 constant salinity at the inflow and another constant salinity at the outflow from the box. For the Arctic Ocean, that is clearly not given. To me it is not clear from this manuscript or from Bacon et al 2015 whether Sbar is an area mean or a transport weighted mean salinity over the boundary. I would appreciate it if the authors could clarify this here.

Our original intention was to avoid over-complication, but again, we agree with the reviewer's sentiment, so as part of the new text in response to the preceding comment, we have revised section 2.4 accordingly :

**P14 L 8** where the integral is taken around the ocean boundary, from seabed to surface, and including sea ice; the overbar indicates area mean and prime indicates deviation from the mean (and following text).

p512-3 and 16-7 "accurate estimates of freshwater flux require the definition of an appropriate reference salinity (Sbar)" and "the boundary-mean salinity is the only appropriate reference salinity" I do not think

that either of these sentences is correct. But rather than arguing over whether they are correct, I would suggest to leave them out as they are in fact not crucial to anything that follows later in the manuscript.

We thought about this, and yes, we agree. Material discussing appropriate "reference salinities" has been removed. The text now appears as below.

**P4 L14** The second way to estimate F is what Aagaard and Carmack (1989) call the "indirect" approach, which we call the "budget" approach. The budget approach recognises that ocean salinity is sensitive to dilution (or concentration) by addition (or removal) of freshwater. Therefore with knowledge of fields of velocity and salinity around the boundary of a closed volume (to ensure conservation of mass), the surface freshwater flux within the volume may be calculated; see (Serreze et al., 2006; Dickson et al., 2007; Bacon et al., 2015).

p5l29 Nd isotopes and REEs have also been used as conservative tracers of different rivers in the Arctic Ocean, e.g. doi:10.1016/j.gca.2016.12.028

An extremely interesting and useful pointer to material of which we were unaware; thank you. The discussion (Section 4) of the manuscript has been expanded to include the use of exotic tracers:

**P26 L10** Nevertheless, other, more exotic species, may prove useful. For instance, Laukert et al. (2017) show that the distri bution of neodymium isotopes in Fram Strait that bears a considerable resemblance to our

"Pacific" water distribution (our Fig. 9, their Fig. 3), and with a similar interpretion to ours (Section 4.2 above) as to the provenance of the water mass. Furthermore, Wefing et al. (2019) analyse isotopes of iodine and uranium, sourced from UK and French nuclear reprocessing plants, which trace Arctic Ocean circulation pathways and residence times, showing that some fraction of the near-surface freshened oceanic waters in the west of Fram Strait, which appear to be of Pacific origin from the N:P analysis, may actually have originated from the Norwegian Coastal Current

p6l14-17 Again, while this is an appropriate step to take at this point (and better results might not be obtained from data at this point), it should still be pointed out that this is not perfect and there are in fact possibly large systematic errors arising from the sampling locations and/or spatially (potentially) insufficient sampling. It would be nice to mention these points with at least a few sentences.

We have deleted this text as part of our overhaul.

*p7l8 The correct statement would be that this "conserves volume and salt transports", not that it "conserves volume and salinity transports"!*

No, this is wrong. The measured property that is transported is called *salinity*. "Salt" in this context is a colloquialism, however commonly it may be used. If you want to check on the history and background, then Bacon et al. (JAOT 2007, 10.1175/JTECH2081.1) give a reasonable overview in the first two sections, and that paper is usefully supplemented by McDougall (Ocean Sci. 2012, 10.5194/os-8-1123-2012) on 'absolute salinity'.

*p7l14 Please say what you mean by the plus/minus here. E.g. it could be standard deviation or standard error.*

How we define the +/- has now been included :

**p7 L21 (\$\pm\$ standard deviation)**

p7l15 "1.0 +- 0.2" not "1 +- 0.2"

p7l18 "Sv" not "sV"

Both errors have now been corrected as indicated.

*p7l24 Please state where your information on sea ice export is from, e.g. satellite observations of sea ice drift and sea ice volume?*

The terms reported here are outputs from the TB12 model. How the sea ice flux was initialised in the model is detailed in TB12 (see para. 39), but in brief, they used remote-sensed area flux (due to Ron Kwok) in combination with thickness flux (due to Edmond Hansen).

We edit our text to:

**P7 L29** The net surface freshwater flux (both liquid and solid) calculated by TB12 is  $187 \pm 44$  mSv, manifest as  $147 \pm 42$  mSv in the liquid ocean plus  $40 \pm 14$  mSv in sea ice.

p8l5 "nutrient and delta18O data were optimally interpolated" Comment on whether the spatial distribution of the data was sufficient or whether there could be interpolation issues.

Reviewer 1 asked a similar question and we refer to our reply above.

p819 "grid cells as hydrographic stations" I see where this is coming from, but it is still a strange way to formulate it.

The statement has now been clarified:

**P8 L16** Our domain comprises a total of 147 hydrographic stations, which includes data from 16 general circulation model grid cells in the Barents Sea Opening that are used as hydrographic stations, covering a total oceanic distance of 1803 km, with a total (vertical) section area of 1050 km2.

p8l15-27 For me this was totally incomprehensible upon first reading. The terms "3EM", "4EM" and "4EM+" are not self explanatory. I would strongly advise to make a diagram or a table. A suggestion would be a table like this (columns could not be formatted in plain text, so the individual lines of the table are *grouped together*): Model name Constraints End members that are solved for Comments new line 3EM Volume conservation, salinity data, delta180 data Seawater fraction, meteoric water fraction, ice melt water fraction Seawater is water with S=35 irrespective of whether it enters from the Atlantic or the Pacific new line 4EM

Volume conservation, salinity data, delta18O data, P\* data Atlantic water fraction, Pacific water fraction, meteoric water fraction, ice melt water fraction Atlantic water and Pacific water are defined to have identical S and delta18O end member characteristics, but different P\* new line 4EM+ Volume conservation, salinity data, delta18O data, P\* data Atlantic water fraction, Pacific water fraction, meteoric water fraction, ice melt water fraction Atlantic water and Pacific water are defined to have different P\* and similar, but not identical S and delta18O end member characteristics

This is a sensible suggestion, so we have added a new Table 1, appropriately referenced in the manuscript text, as a compact display of the three model schema.

I would also already add a sentence like the following one here, because (anyways for me) it was not clear why you do these two versions with 4EM and 4EM+: "4EM+ is degenerate (meaning that numerical values are strongly affected by small perturbations) because the distinct source salinity and delta18O values of Pacific water are on a mixing line between the meteoric and Atlantic Water end member quantities.

We include the 4EM+ schema as despite it being degenerate, this represents what is becoming common practice in geochemical tracer studies as noted in the text :

**P9 L28** Thirdly the 4EM scheme is applied again, but now adopting distinct end-member properties for both ocean-source salinity and  $\delta$  18 O (4EM+), replicating previous practice (Dodd et al., 2012; Jones et al., 2008; Sutherland et al., 2009). The properties of the three schemes are summarised in Table 1.

Comments about the degeneracy of the 4EM+ scheme are made in the discussion (see point below).

p9l14 1 sentence here why you use Pest: in order to judge the method, not for use in the method itself

Pest (now P\_oce) is used to calculate end-member values of P\*. The text has now been updated to make this clearer:

**P10 L17** where P\_oce is the estimated concentrations of phosphate from the relevant ocean (either Atlantic and Pacific) waters and the subscripts slope and int indicate the slope and intercept of the relationships.

p10l5 Again, I think this would be much clearer if you could refer to the table as I suggested above.

A reference to the table is now included.

p11l14 refer to Table 2 in this sentence

A reference to the table is now included.

p12l16 Again, area mean or transport weighted mean?

This has now been clarified in the text:

**P14 L5** Seawater salinity for 3EM and 4EM models is fixed at the boundary area mean salinity for the TB12 model (34.662).

**p14l1 m/s (see my comment below on Figure 6)**

The units of Sv are correct - in all cases we take a volume flux from the TB12 model, which is calculated as a velocity (v) \* grid cell area (horizontal distance ds x vertical distance dz) and scale it by the estimated water type fraction.

p14l11 It might be helpful to remind the reader that your +- values stem from the Monte Carlo simulations.

A reminder has been now included in the text :

**P16 L2** For the 3EM model schemes, the net seawater volume flux is effectively zero ( $0.002 \pm 0.006$  Sv, Table 4, Monte Carlo uncertainty quantification).

p15l2 Also, here 1 sentence would be in order repeating what the difference between 4EM and 4EM+ is and why you do both calculations.

We have revised the text as suggested :

**P16 L11** The 4EM scheme extends the 3EM scheme through use of inorganic nutrient (nitrate and phosphate) data, aiming to discriminate between Atlantic and Pacific seawater origin. The 4EM scheme retains single end-points for salinity and  $\delta^{18}$ O, as in 3EM. In the 4EM+ scheme, distinct salinity and  $\delta^{18}$ O end-member properties are attributed to Atlantic and Pacific seawaters, replicating previous practice (Dodd et al., 2012; Jones et al., 2008; Sutherland et al., 2009).

p15l22-23 and l25-26 Please don't just show both sets of numbers, but also comment on which one you think makes more sense.

This text is no longer in the manuscript.

p16l16 Add a sentence such as: "Both of these numbers should be approximately 0 and therefore, we consider this a model/methodological/data(?) mistake for the following reasons..." and

p16l21 You are only looking at data from 1 summer month. Discuss whether all of this should be balanced in the quasi-synoptic view of the data you use.

This text is no longer in the manuscript.

p17l16-18 Neither of these views seems plausible for the West Spitsbergen Current. Should the Atlantic water salinity not rather match the WSC closely?

We have added a new and detailed discussion of the WSC attribution to the manuscript, in Section 4.1 on ice-modified waters, starting as below.

**P20 L1** The models generate apparent brine imports in the WSC and the Barents Sea Opening, both of magnitude ~ 45 mSv, a total of ~ 90 mSv with a large relative uncertainty of ~ 50 mSv. If correct, this is a substantial component of the Arctic Ocean freshwater budget. These (apparent) fluxes are too small to be visible on Fig. 5, but for scale, note that each new (oceanic water) inflow is ~ 3 Sv, 1 % of which is 30 mSv.

These brine fluxes are consequences of weakly positive  $\delta^{18}$ O anomalies centred around ~ 300 m depth in both locations, each about 200 m thick and each spanning ~ 200 km. The presence of these features in both Fram Strait and the Barents Sea Opening suggests that they are source water (Atlantic seawater) properties and not the result of modifications by local processes. Frew et al. (2000) examine the oxygen isotope composition of northern North Atlantic water masses from measurements made in 1991. Considering the waters of interest here – the upper ~ 500 m in the eastern North Atlantic (their stations 10, 24, 26, 72) – we find (broadly) salinities and  $\delta^{18}$ O values in the ranges 35.0 – 35.2 and 0.2 – 0.4 ‰ respectively (their Fig. 2). This combination and range describes the part of the dense cloud of points heading a short distance "north-eastwards" in phase space away from the seawater endpoint (Fig. 3 panel a inset). A consistent interpretation of the apparent WSC and Barents Sea Opening brine imports, therefore, is that they are actually manifestations not of local processes but rather of source water variability, in the light of our salinity (34.662) and  $\delta^{18}$ O (mean 0.2 ‰) endpoints.

p18l14-15 Should this not be considered everywhere?

This text is no longer in the manuscript.

p19l2 "solid (sea ice) fraction" instead of "solid, sea ice, fraction"

The discussion has been substantially updated and these lines are no longer present.

p19l27 "(at least when considering full depth assessments)" It is not clear why that caveat is necessary and why the sentence is not correct without the added information in brackets.

This text is no longer in the manuscript.

p2016 How can a river be a sink? Processes on the continental shelf near the river could be sink processes.

This text is no longer in the manuscript. The discussion of nutrients in Section 4.2 is substantially re-cast.

p20l20 no ":"

This text is no longer in the manuscript.

p20l25 Explain how I would see that from Figure 3 and what degenerate means in that context.

We discuss degeneracy now at the end of Section 4.2 (P24 L10), in the new Section 4.3 (P25 L1-13), and Section 4.4 (P26 L4-7).

p2111 "boundary mean salinity" Again, where do I "see" that in Figure 3?

This text is no longer in the manuscript.

p21112-24 This text and the associated Figures 11-13 should in my opinion be removed from the manuscript as it is unclear what you mean by "oceanic origin freshwater". Additionally, there is no insightful information contained in them.

We retain Figure 11, since it is a useful visualisation, but otherwise, yes, you are right, so we have removed Figures 12 & 13. The "oceanic freshwater" comment was removed as part of our clean-up of terminology mentioned in our introductory comments to these replies.

p22l17 "Carmack et al. (2016, Appendix)"

This text is no longer in the manuscript.

p2316 "salt conservation"!

This text is no longer in the manuscript.

Tab1 Why is there a larger line break after the first line of delta180 and salinity?

Table format has been corrected.

Tab1 2nd line under ice melt: What is "surf"?

The table has been updated and the caption expanded to clarify this point:

**Table 2 caption** End-member values for salinity and  $\delta^{18}O$  (‰) from the literature. Note Bauch et al. (1995) calculate ice melt  $\delta^{18}O$  by multiplying measured surface seawater  $\delta^{18}O$  (surf ) by a "fractionation factor" of 1.0021.

Tab3 Similar to p7l24, where is the information about -0.040Sv solid ice melt from?

The caption has now been updated to make this clear :

Table 4 caption Values of solid freshwater flux from Tsubouchi et al. (2012).

Fig2 I think the other piece of interpolated data that your study is based on is cross-sectional velocity. I would recommend to add this as a top (4th) panel to Fig2. In that case the reader does not need to refer back to TB12 to get that information

We haven't done this because it would be the same as the volume transport plot (panel d), apart from scale.

Fig2 caption l1 "P\*" should be with a superscripted "\*"

Now corrected.

Fig2 caption 14 Repeat what the main Arctic water masses are so that the reader does not need to refer back to TB12.

and

*Fig2 caption Add: "Note the broken scaling of the y-axis."*

Definitions of the Arctic water masses from TB12 have now been included in the figure caption as has the comment about the y-axis scale.

**Fig 2 Caption** Sections of  $\delta^{18}$ O (panel a), salinity (panel b), P\* (panel c) and volume flux from Tsubouchi et al. (2012)(panel d) after optimal interpolation onto the Tsubouchi et al. (2012) CTD station positions, clockwise around the four gateways from Davis to Bering Straits. Solid black lines indicate the potential density ( $\sigma$ ) surfaces separating the main Arctic water masses grouped as follows, surface water ( $\sigma$ 0< 26.0), subsurface water ( $26.0 < \sigma$ 0 < 27.1), upper Atlantic water ( $27.1 < \sigma$ 0 < 27.5), Atlantic water ( $\sigma$ 0=27.5 to

 $\sigma$ 0.5=30.28), intermediate water ( $\sigma$ 0.5=30.28 to  $\sigma$ 1=32.75), and deep water ( $\sigma$ 1 > 32.75); definitions from Tsubouchi et al. (2012). Note the broken scaling of the y-axis.

Fig3 Your 3EM model solves the classical end member decomposition in the triangle that is drawn in panel a. Your 4EM models essentially are the same, only that they solve the end member decomposition in the tetrahedron that would result if you were to extend panel a in the vertical with the vertical axis being P\*. Since you can't add a 3 dimensional figure to the paper, I would recommend to at least add plane views of this tetrahedron with the data and dashed lines plotted into the panels just as you are doing in panel a right now. Common axes can be aligned with each other. My suggestion: 4 panel figure. top left panel as your panel a. top right panel x-axis P\* y-axis delta18O, bottom left panel x-axis salinity y-axis P\*, bottom right panel your current panel b. Also please substitute the current legend in panel b by a legend for the dashed lines and comment in the figure caption that all symbols and lines are the same in all panels. The 18 in the ylabel of panel a should be superscripted not subscripted. and

Fig3 caption l4 "Dashed thick"

We think it would over-complicate to attempt to graphically reproduce in 2 dimensions a 3-dimensional phase space; we have made the other corrections have now been made as indicated.

**Fig 3 caption** Panel a: Salinity -  $\delta$  18 O relationship for all samples used in this manuscript; mean literature end-points (± standard deviation) are marked. Red crosses indicate the mean values of literature end-points and black dashed lines the mixing lines between them. Panel b: Nutrient data for all samples used in this manuscript compared to the published N:P relationships of Jones et al. (2008), Dodd et al. (2012), Sutherland et al. (2009). The dashed red line indicates a best fit to the Bering Strait nutrient data presented here. Symbols denoting the data from each section are common to both panels. Note Dodd et al. (2012) uses the same Pacific relationship as Jones et al. (2008).

Your units in Figs 6/8/10 and 11a/b are wrong. They should be "Sv/m/km" or more conventionally "m/s". Note that you only arrive at units of transport (Sv) after integrating the data in the figures in the horizontal and vertical dimensions. Same applies for Figs 12/13 where your units should be m2/s or Sv/km or similar.

We apologise sincerely for an error here (application of mistaken scaling). What we should have plotted was indeed volume transports (gridded v x area), but the units should have been mSv and the range more like  $\pm 20$  mSv. This has been corrected.

Fig7 What is plotted in panels 7a and 7b is different from what is plotted in panels 5a and 5b, yet the values in the Met. and Ice Melt columns of Tables 3 and 5 are identical. In my opinion, only either one of those can be correct.

This was a problem with contour levels in the figures. The tables are, in all cases, definitive. All figures have all been re-contoured (all corresponding panels use the same level boundaries for ease of comparison).

Fig11/12/13 can in my opinion be deleted from the manuscript. One of the reasons is that I do not understand what the black line in Fig12/13 is supposed to be.

As noted above, we largely agree.

The legend has now been updated as suggested.

p48l14-16 A correct reference to this data publisher contains the complete DOI and it is not a tech report. Compare how the citation is provided on the webpage of the data set. In addition, you have the wrong title which means that it took me some time to find the data set you are referring to! "Kattner, Gerhard (2011): Inorganic nutrients measured on water bottle samples during POLARSTERN cruise ARK-XXI/1. PANGAEA, https://doi.org/10.1594/PANGAEA.761684"

Thank you very much for the updated reference. This has now been added.

**Short Comment (PD)**

We note first that we have substantially altered our text around these issues: see the new Section 4.2 on "Pacific" water that elaborates on our responses below.

My first comment regards the implementation of the N:P ratio method used to identify Pacific Water. Figure 7 in your paper shows low fractions (10 - 20 %) of Pacific Water along much of the boundary section in places that seem unlikely. For example, much of Fram Strait is filled with low fractions of Pacific Water below 1000 m. Pacific Water is buoyant and enters the stratified Arctic through a 60 m deep channel, so it seems unlikely that Pacific Water should be found at the bottom of Fram Strait. I think these apparently-spurious Pacific Water fractions might need be addressed before we can expect good results from the inverse model.

The N:P ratio method has been around for over twenty years now, and depends on the perception that Atlantic and Pacific source waters occupy distinct locations in nitrate:phosphate phase space. Measurements cluster around two lines with similar slopes, and where the Atlantic-origin waters are offset relative to Pacific waters: for a given phosphate concentration, Pacific nitrate concentration is lower than Atlantic. The offset is ~10 µmol-N / kg.

We contend that, while the source-water attribution is uncontentious, the product-water attribution is suspect. We accept, of course, that much is presently unknown with regard to Arctic biochemical nutrient cycling, and we (or rather, some of us) listed the major issues in the conclusions to Torres-Valdes et al. (GRL 2016). Denitrification is a key process in the Arctic that converts nitrate to  $N_2$ , where it is lost to the system. Chang & Devol (DSR 2009) examine Arctic denitrification rates, finding total rates in the broad range 14-66 kmol-N / s. They find that denitrification occurs mainly in two areas, the Chukchi Sea (for 26% of the total) and the Barents Sea (for 43% of the total).

Consider now, therefore, how denitrification might "convert" Atlantic water into "Pacific" water, by removing nitrate at the observed offset rate of 10  $\mu$ mol / kg. Take (for scale) 1 Sv of Atlantic water; that is 106 m3 / s, or 109 kg / s. Then the required nitrate removal rate is 10  $\mu$ mol / kg x 109 kg / s = 10 kmol / s, which is well within the limits of the Chang & Devol estimate. So it is actually easy to imagine that some "Pacific" water export might actually have originated as Atlantic water, which now carries a denitrification signal.

This hypothesis gives a further clue as to the reason for the presence of low concentrations of "Pacific" water below 1000 m in Fram Strait. Dense water formation in the Arctic is difficult to observe, but given that the lowest deep and bottom water temperatures occur in the Nansen and Amundsen Basins, and their likely origin through winter-time dense water formation is the Barents Sea, it is reasonable to suppose that denitrification of Atlantic waters also explains the sub-1000 m presence of "Pacific" water. This view is further supported by the Laukert et al. view of neodymium isotopes in Fram Strait.

The recently-published Alkire et al. (GRL 2019) note in their Introduction ways in which traditional identification of Pacific-origin seawater, via silicate concentrations and nitrate:phosphate ratios, may be growing unreliable as reduction in sea ice concentrations over the East Siberian Sea has enabled interactions with sediments leading to production of Halocline waters that are geochemically similar to Pacific waters.

There are a couple of ways in which this might be achieved:

1) Some of the apparently-spurious low Pacific Water fractions might arise from uncertainties in the endmember properties. If the low fractions are not significantly different from zero it might be justifiable to suppress them.

The concentrations are low but still significantly greater than zero; they are not explained by endpoint uncertainty.

2) An alternative approach could be to apply the N:P technique only in the depth range where Pacific Water is likely to be found, assuming fractions below some depth threshold to be zero. The N:P ratio method has a large errors associated with it and if it is applied indiscriminately over large areas where we would not expect to find Pacific Water the accumulated systematic errors probably become quite significant.

Considering the wider Arctic Mediterranean, extending to include the Nordic Seas southwards to the Greenland-Scotland Ridge, there is no exchange with the wider world deeper than 800 m (in the Faroe Bank Channel). The circulation in Fram Strait below 1000 m is in near-balance (transport northwards is nearly equal to transport southwards). Combined with the near-uniform distribution of "Pacific" fraction, it has a negligible impact on "Pacific" water fluxes. We checked this by repeating our calculations excluding the sub-1000 m layers in Fram Strait, and we still find ~1 Sv excess of "Pacific" water: (roughly) 1 Sv in through Bering Strait, 2 Sv out, mainly through Davis Strait.

However, the limitations of the N:P ratio technique are perhaps not the main reason that the inverse model does not balance for Pacific Water. My second comment regards the application of the inverse technique to Pacific Water in the Arctic Ocean. I'm not very familiar with inverse modelling, but I think the technique assumes that the system is in a steady state. The repeated Pacific Water sections in Dodd et al., 2012 (cited in your paper) indicate that the flow of Pacific Water through the Arctic is not in a steady state. At least in Fram Strait, Pacific Water is released in pulses with peak Pacific Water fractions of up to 80 % interspaced with periods where peak Pacific Water fractions barely exceed 20 %. The duration of pulses is probably of the order of 2 years, which is quite short relative to the time required for Pacific water to cross the Arctic. I'm not exactly sure how this can be best addressed, but I think the paper should at least discuss this steady-state issue.

This is an interesting point. As you rightly say, Dodd et al. (2012) shows high variability in "Pacific" water fraction in Fram Strait. However, what we learn both from that paper and from Torres-Valdes et al. (JGR 2013) as well as from our present manuscript is that Fram Strait is a minority contributor to net "Pacific" water export, so that what Dodd et al. present in Fram Strait is actually low variability around a low mean, leading to high relative variability. The Davis Strait "Pacific" water export is dominant, and there is no version of our calculation that can significantly reduce the mismatch between the 1 Sv Pacific water import and the 2 Sv "Pacific" water export. Using, for example, the 1998 Fram Strait section with its high concentration of "Pacific" water would only increase further the net "Pacific" water export rate.

One reason that the inverse model might balance for salinity/freshwater, but not for Pacific Water, could be that in years when Pacific Water is not present in a given location it tends to be replaced by another halocline water mass of similar density (ie: rather similar salinity).

There is some denitrification in the Arctic and I agree that when using the N:P ratio technique, some Atlantic Water will apparently be transformed into Pacific Water over the shallow shelves. That is indeed a fundamental limitation of the technique. However, if the steady-state issue is as serious as I think it is, then I'm not sure that the results of the inverse model give us much new information about the reliability of the N:P technique. Please do correct me if I am wrong about something here though!

As a final point, Alkire et al. (GRL 2019) use the quasi-conservative tracer "NO", as well as dynamic height, to examine the front separating Pacific and Atlantic halocline waters in the East Siberian Sea. They find that "traditional tracers", meaning the N:P ratio, "used to quantify Pacific water contributions to the Arctic Ocean are no longer accurate". There is no combination of transport uncertainties (as Tsubouchi et al. 2012) with inorganic nutrient concentration uncertainties (Torres-Valdes et al. 2013) that can more than double the apparent Pacific water flux, from ~1 Sv to ~2.5 Sv.

[revised manuscript text omitted]

- 25 freshwater from seawater representing net by evaporation and sea ice formation, respectively. Seawater However, seawater fractions, either total or individual Atlantic and Pacific

water fractions, should be positive. Consequently, Pacific and Atlantic water fractions were made positive-definite by rounding to zero any of the fractions that were less than zero, and setting the remaining seawater fraction so that equation 1 was not invalidated.

**3.1 Three end-member model (3EM)**

- 5 The distribution of 3EM source fractions is shown in Fig. 5. Ice-modified waters are found almost exclusively in the surface / upper waters of the model (depths down to 1000 dbar in the Davis Strait), with highest-magnitude fractions (-0.15) found in sub-surface waters of the western Fram Strait between depths of  $\sim$  50 and 300 dbar. The fractions of ice-modified waters are mostly negative(indicating high salinity from upstream ice formation),
- 10 indicating brine, with a small fraction (~ 0.05) positive (indicating fresh meltwatermelt water input) in the surface (above 70 dbar) East Greenland Current (EGCEast Greenland Current; between 6.5 and  $2^{\circ}$  W) of the Fram Strait. Meteoric waters are also found almost exclusively in the surface / upper waters of the model, with high fractions (> 0.08) in the surface / subsurface waters (depths down to 350 dbar) in the Davis Strait and the western side of the
- $_{15}$  Fram Strait. There is also a high fraction of meteoric water in the Bering Strait. Seawater fractions are high ( $\sim$  1) in all deep / intermediate model waters at depths in excess of  $\sim$  350 dbar.

Typical volume fluxes (positive indicating into the Arctic) for the 3EM source fractions are shown in Fig. 6. The strongest fluxes of ice-modified waters occur as brine exports in

- 20 surface waters of the middle of the Davis Strait , and on the western side of the Fram Strait (EGC)and East Greenland Current), and as brine import to the east in the Bering Strait, with fluxes of ~ 0.1 Sv in magnitude(positive fluxes indicating an export of high-salinity waters). The patterns of countervailing fluxes over the Belgica Bank (west of 6.5° W) in the Fram Strait are indicative of indicate recirculation (see TB12). Meteoric water volume fluxes
- follow the same general pattern as for ice-modified waters, with strong export (~ 0.1 Sv) in the middle of the Davis Strait and the EGC. There is a East Greenland Current and strong import (~ 0.1 Sv) of meteoric waters in the Bering Strait. Seawater volume fluxes indicate a strong export mid-Davis Strait (~ 1 resemble the oceanic circulation of TB12 (as expected),

with moderate export in the EGC concentrated exports in Davis Strait (~ 1 Sv)and the East Greenland Current (~ 0.5 Sv)and moderate import, 
[revised manuscript text omitted]

|            | Volume                                                                                                                                                                                                    | Atlantic seawater,                                                     | Atlantic and Pacific                                                                      |
| 4EM | conservation,
salinity, $\delta^{18}O$ ,
$P_{*}^{*}$                                                                                                                                                | Pacific seawater,
meteoric water, ice
melt                       | seawaters are assigned a common salinity and $\delta^{18}O$ , but different $P^*$ values. |
| 4€M+       | $\begin{array}{c} \underbrace{ \text{Volume}} \\ \underbrace{ \text{conservation,}} \\ \underbrace{ \text{salinity,}} \\ \underbrace{ P^* \\ \end{array} \end{array} \overset{\delta^{18}O}, \end{array}$ | Atlantic seawater,
Pacific seawater,
meteoric water, ice
melt | AtlanticandPacificseawatershavedifferentsalinity, $\delta^{18}O$ and $P^*$ values.        |

 Table
 I. Description of the three model schemes.

|                       | Atlantic                                                                    | Pacific                                                  | mean
Met.                      | Ice Melt                                      | Source                                                                                              |
|-----------------------|-----------------------------------------------------------------------------|----------------------------------------------------------|------------------------------------------|-----------------------------------------------|-----------------------------------------------------------------------------------------------------|
| $\delta^{18}O$        | $0.24\pm0.03$                                                               | $-0.8\pm0.1$                                             | $-20\pm2$                                | $-2\pm1.0$                                    | Yamamoto-Kawai et al. (2008)                                                                        |
| <del>(‰)</del>
(∞) | $\begin{array}{c} 0.3 \\ 0.3 \\ 0.19 \pm 0.06 \\ 0.35 \pm 0.15 \end{array}$ | $-1.0 \pm 0.5$
-1.3
$-0.8 \pm 0.1$
$-1 \pm 0.1$ | $-21 \\ -18.4 \\ -18 \pm 2 \\ -21 \pm 2$ | $\frac{surf + 2.1}{0.5}$
-2 ± 1
1 ± 0.5 | Bauchaet/al. (1995)
Dodd et al. (2012)
Azetsu-Scott et al. (2012)
Sutherland et al. (2009) |
| Mean                  | 0.28                                                                        | -0.98                                                    | -19.7                                    | -0.6                                   |                                                                                                     |
| Sal. <del>(PSI</del>  | J }34.87 ± 0.03                                                      | $32.5\pm0.2$                                             | 0                                        | $4\pm1$                                       | Yamamoto-Kawai et al. (2008)                                                                        |
| (PSU)          | 34.92                                                                       | 33                                                       | 0                                        | 3                                             | Bauch et al. (1995)                                                                                 |
|                       | 34.9                                                                        | mean 32.0                                         | 0                                        | 4                                             | Dodd et al. (2012)                                                                                  |
|                       | $34.75\pm0.14$                                                              | $32.5\pm0.2$                                             | 0                                        | $4\pm1$                                       | Azetsu-Scott et al. (2012)                                                                          |
|                       | $35 \pm 0.15$                                                               | $32.7\pm1$                                               | 0                                        | $4\pm1$                                       | Sutherland et al. (2009)                                                                            |
| Mean                  | 34.89                                                                       | 32.54                                                    | 0                                        | 3.75                                          |                                                                                                     |

**Table 2.** End-member values for salinity and  $\delta^{18}O$  (‰) from the literature. Note Bauch et al. (1995) calculate ice melt  $\delta^{18}O$  by multiplying measured surface seawater  $\delta^{18}O$  (*sur f*) by a "fractionation factor" of 1.0021.

5

|          | Slope           | Intercept                       | Source                                      |
|----------|-----------------|---------------------------------|---------------------------------------------|
| Atlantic | 0.0545          | 0.1915                          | Jones et al. (2008)                         |
|          | 0.053           | 0.170                           | Dodd et al. (2012)                          |
|          | $0.048\pm0.003$ | $0.130\pm0.04$                  | Sutherland et al. (2009)                    |
| Mean     | 0.052           | 0.164                           |                                             |
| Pacific  | 0.0653          | 0.94                            | Jones et al. (2008)                         |
|          | $0.08\pm0.015$  | $\textbf{0.85}\pm\textbf{0.13}$ | Sutherland et al. (2009)                    |
|          | 0.0654          | 0.6766                          | Calculated for this study from observations |
| Mean     | 0.070           | 0.822                           |                                             |

Table 3. P:N relationships, where  $PO_4 = Slope * NO_3 + Intercept \ (\mu \text{ mol kg}^{-1})$

|         | Oceanic            | Met.                                | Ice Melt                             | Sum    |
|---------|--------------------|-------------------------------------|--------------------------------------|--------|
| Davis   | $-3.035 \pm 0.008$ | -0.209 $\pm$ 0.055                  | $0.100\pm0.062$                      | -3.144 |
| Fram    | -1.566 $\pm$ 0.004 | -0.104 $\pm$ 0.027                  | $0.038\pm0.030$                      | -1.632 |
| Barents | $3.671 \pm 0.004$  | $\textbf{0.013} \pm \textbf{0.031}$ | $\textbf{-0.048} \pm \textbf{0.035}$ | 3.636  |
| Bering  | $0.931\pm0.003$    | $0.099 \pm 0.023$                   | -0.029 $\pm$ 0.026                   | 1.001  |
| Liquid  | $0.002\pm0.006$    | $-0.200 \pm 0.044$                  | $0.060\pm0.050$                      | -0.139 |
| Solid   |                    |                                     | $\textbf{-0.040} \pm \textbf{0.014}$ | -0.04  |

**Table 4.** Mean volume fluxes (Sv  $\pm$  standard deviation) for the three end-member (3EM) model. Positive values indicate fluxes into the Arctic. Values of solid freshwater flux from Tsubouchi et al. (2012).

|        | Oceanic                             | Met.                                 | Ice Melt                             | Sum    |
|--------|-------------------------------------|--------------------------------------|--------------------------------------|--------|
| BB     | $-0.350 \pm 0.001$                  | -0.022 $\pm$ 0.006                   | -0.002 $\pm$ 0.006                   | -0.373 |
| EGC    | -5.364 $\pm$ 0.007                  | $\textbf{-0.083} \pm \textbf{0.050}$ | $0.088\pm0.056$                      | -5.359 |
| Mid.   | $0.303\pm0.000$                     | $\textbf{-0.000} \pm \textbf{0.003}$ | $\textbf{-0.005} \pm \textbf{0.003}$ | 0.298  |
| WSC    | $\textbf{3.845} \pm \textbf{0.004}$ | $0.001\pm0.032$                      | $\textbf{-0.044} \pm \textbf{0.036}$ | 3.803  |
| Liquid | $-1.566 \pm 0.004$                  | $-0.104 \pm 0.027$                   | $0.038\pm0.030$                      | -1.632 |

**Table 5.** Mean volume fluxes (Sv  $\pm$  standard deviation) for the components of the Fram Strait flux(Belgica Bank, BB; East Greenland Current, EGC; Mid-strait, Mid.; West Spitsbergen Current, WSC)from the three end-member (3EM) model. Positive values indicate fluxes into the Arctic.

|         | Atlantic                             | Pacific                              | Met.                                 | Ice Melt                             | Sum    |
|---------|--------------------------------------|--------------------------------------|--------------------------------------|--------------------------------------|--------|
| Davis   | $\textbf{-0.815} \pm \textbf{0.346}$ | $\textbf{-2.219} \pm \textbf{0.346}$ | -0.209 $\pm$ 0.055                   | $0.100\pm0.062$                      | -3.144 |
| Fram    | $\textbf{-1.333} \pm \textbf{0.088}$ | $\textbf{-0.233} \pm \textbf{0.088}$ | -0.104 $\pm$ 0.027                   | $\textbf{0.038} \pm \textbf{0.030}$  | -1.632 |
| Barents | $\textbf{3.520} \pm \textbf{0.184}$  | $0.151\pm0.184$                      | $\textbf{0.013} \pm \textbf{0.031}$  | $\textbf{-0.048} \pm \textbf{0.035}$ | 3.636  |
| Bering  | $0.126\pm0.076$                      | $0.806\pm0.076$                      | $0.099 \pm 0.023$                    | -0.029 $\pm$ 0.026                   | 1.001  |
| Liquid  | $1.497\pm0.268$                      | $\textbf{-1.495} \pm \textbf{0.268}$ | $\textbf{-0.200} \pm \textbf{0.044}$ | $0.060\pm0.050$                      | -0.139 |
| Solid   |                                      |                                      |                                      | $\textbf{-0.040} \pm \textbf{0.014}$ | -0.04  |

**Table 6.** Mean volume fluxes (Sv  $\pm$  standard deviation) for the four end-member (4EM) model. Positive values indicate fluxes into the Arctic. Values of solid freshwater flux from Tsubouchi et al. (2012).

|        | Atlantic                             | Pacific                             | Met.                                 | Ice Melt                             | Sum    |   |
|--------|--------------------------------------|-------------------------------------|--------------------------------------|--------------------------------------|--------|---|
| BB     | $\textbf{-0.182} \pm 0.035$          | -0.167 $\pm$ 0.035                  | $\textbf{-0.022}\pm0.006$            | -0.002 $\pm$ 0.006                   | -0.373 |   |
| EGC    | $\textbf{-4.948} \pm \textbf{0.376}$ | -0.416 $\pm$ 0.377                  | $\textbf{-0.083} \pm \textbf{0.050}$ | $\textbf{0.088} \pm \textbf{0.056}$  | -5.359 |   |
| Mid.   | $0.226\pm0.058$                      | $0.077\pm0.058$                     | $\textbf{-0.000} \pm \textbf{0.003}$ | $\textbf{-0.005} \pm \textbf{0.003}$ | 0.298  |   |
| WSC    | $\textbf{3.571} \pm \textbf{0.274}$  | $\textbf{0.274} \pm \textbf{0.275}$ | $0.001\pm0.032$                      | $\textbf{-0.044} \pm \textbf{0.036}$ | 3.803  |   |
| Liquid | $-1.333 \pm 0.088$                   | $-0.233 \pm 0.088$                  | $-0.104 \pm 0.027$                   | $0.038\pm0.030$                      | -1.632 | _ |

**Table 7.** Mean volume fluxes (Sv  $\pm$  standard deviation) for the components of the Fram Strait flux (Belgica Bank, BB; East Greenland Current, EGC; Mid-strait, Mid.; West Spitsbergen Current, WSC) from the four end-member (4EM) model. Positive values indicate fluxes into the Arctic.

|         | Atlantic                             | Pacific                              | Met.                                 | Ice Melt                             | Sum    |
|---------|--------------------------------------|--------------------------------------|--------------------------------------|--------------------------------------|--------|
| Davis   | -0.934 $\pm$ 0.343                   | $-2.231 \pm 0.367$                   | $\textbf{-0.060} \pm \textbf{0.057}$ | $0.080\pm0.084$                      | -3.144 |
| Fram    | $\textbf{-1.333} \pm \textbf{0.079}$ | $\textbf{-0.234} \pm \textbf{0.086}$ | -0.091 $\pm$ 0.025                   | $0.026\pm0.030$                      | -1.632 |
| Barents | $\textbf{3.493} \pm \textbf{0.168}$  | $0.151\pm0.185$                      | $0.011\pm0.037$                      | $\textbf{-0.019} \pm \textbf{0.050}$ | 3.636  |
| Bering  | $\textbf{0.158} \pm \textbf{0.089}$  | $\textbf{0.825} \pm \textbf{0.099}$  | $0.041\pm0.030$                      | $\textbf{-0.023} \pm \textbf{0.034}$ | 1.001  |
| Liquid  | $1.384\pm0.255$                      | $\textbf{-1.488} \pm \textbf{0.263}$ | $\textbf{-0.098} \pm \textbf{0.046}$ | $0.063\pm0.064$                      | -0.139 |
| Solid   |                                      |                                      |                                      | $\textbf{-0.040} \pm \textbf{0.014}$ | -0.04  |

**Table 8.** Mean volume fluxes (Sv  $\pm$  standard deviation) for the four end-member (4EM+) model. Positive values indicate fluxes into the Arctic. Values of solid freshwater flux from Tsubouchi et al. (2012).

43

|        | Atlantic                            | Pacific            | Met.                                 | Ice Melt                             | Sum    |
|--------|-------------------------------------|--------------------|--------------------------------------|--------------------------------------|--------|
| BB     | -0.191 $\pm$ 0.034                  | -0.167 $\pm$ 0.035 | -0.011 $\pm$ 0.005                   | -0.004 $\pm$ 0.007                   | -0.373 |
| EGC    | -4.929 $\pm$ 0.345                  | -0.416 $\pm$ 0.376 | -0.060 $\pm$ 0.057                   | $0.046\pm0.073$                      | -5.359 |
| Mid.   | $0.231\pm0.053$                     | $0.076\pm0.056$    | $\textbf{-0.007} \pm \textbf{0.004}$ | -0.003 $\pm$ 0.005                   | 0.298  |
| WSC    | $\textbf{3.556} \pm \textbf{0.251}$ | $0.274\pm0.273$    | $\textbf{-0.013} \pm \textbf{0.040}$ | $\textbf{-0.014} \pm \textbf{0.051}$ | 3.803  |
| Liquid | $-1.333 \pm 0.079$                  | $-0.234 \pm 0.086$ | $-0.091 \pm 0.025$                   | $0.026 \pm 0.030$                    | -1.632 |

**Table 9.** Mean volume fluxes (Sv  $\pm$  standard deviation) for the components of the Fram Strait flux (Belgica Bank, BB; East Greenland Current, EGC; Mid-strait, Mid.; West Spitsbergen Current, WSC) from the four end member (4EM+) model. Positive values indicate fluxes into the Arctic.

|         | Oceanic                       | Met.                                 | Ice Melt                             | Sum    |
|---------|-------------------------------|--------------------------------------|--------------------------------------|--------|
| Davis   | $-3.003 \pm 0.007$            | $-0.219 \pm 0.049$                   | $\underline{0.078 \pm 0.055}$        | -3.144 |
| Fram    | $-1.550 \pm 0.003$            | $\textbf{-0.109} \pm \textbf{0.024}$ | $\underline{0.026 \pm 0.027}$        | -1.632 |
| Barents | $3.633\pm0.001$               | $0.025\pm0.007$                      | $\textbf{-0.022} \pm \textbf{0.007}$ | 3.636  |
| Bering  | $\underline{0.921 \pm 0.003}$ | $\underline{0.102\pm0.022}$          | $-0.023 \pm 0.025$                   | 1.001  |
| Liquid  | $0.002\pm0.006$               | $-0.200 \pm 0.044$                   | $\underline{0.060\pm0.050}$          | -0.139 |
| Solid   |                               |                                      | $\textbf{-0.040} \pm \textbf{0.014}$ | -0.04  |

Table 10. Mean volume fluxes (Sv  $\pm$  standard deviation) for a three end-member model with seawater salinity and  $\delta^{18}O$  fixed at 35.0 and 0.35 ‰, respectively. Positive values indicate fluxes into the Arctic. Values of solid freshwater flux from Tsubouchi et al. (2012).
|        | Oceanic                              | Met.               | Ice Melt                             | Sum    |
|--------|--------------------------------------|--------------------|--------------------------------------|--------|
| BB     | $-0.346 \pm 0.001$                   | $-0.023 \pm 0.005$ | $-0.004 \pm 0.006$                   | -0.373 |
| EGC    | $\textbf{-5.309} \pm \textbf{0.003}$ | $-0.100 \pm 0.023$ | $0.050\pm0.026$                      | -5.359 |
| Mid.   | $0.300\pm0.000$                      | $0.001\pm0.000$    | $\textbf{-0.003} \pm \textbf{0.001}$ | 0.298  |
| WSC    | $3.805\pm0.001$                      | $0.014\pm0.004$    | $-0.016 \pm 0.004$                   | 3.803  |
| Liquid | $-1.550 \pm 0.003$                   | $-0.109 \pm 0.024$ | $0.026 \pm 0.027$                    | -1.632 |

**Table 11.** Mean volume fluxes (Sv  $\pm$  standard deviation) for the components of the Fram Strait flux (Belgica Bank, BB; East Greenland Current, EGC; Mid-strait, Mid.; West Spitsbergen Current, WSC) for a three end-member model with seawater salinity and  $\delta^{18}O$  fixed at 35.0 and 0.35 ‰, respectively. Positive values indicate fluxes into the Arctic.

5